# Data Free Backdoor Attacks

**Bochuan Cao[1], Jinyuan Jia[1], Chuxuan Hu[2], Wenbo Guo[3],**
**Zhen Xiang[4], Jinghui Chen[1], Bo Li[2], Dawn Song[5]**

[1]The Pennsylvania State University     [2]University of Illinois at Urbana-Champaign
[3]University of California, Santa Barbara     [4]University of Georgia
[5]University of California Berkeley

bccao@psu.edu, jinyuan@psu.edu, chuxuan3@illinois.edu,
henrygwb@ucsb.edu, zhen.xiang.lance@gmail.com, jzc5917@psu.edu,
lbo@illinois.edu, dawnsong@cs.berkeley.edu

## Abstract

Backdoor attacks aim to inject a backdoor into a classifier such that it predicts any input with an attacker-chosen backdoor trigger as an attacker-chosen target class. Existing backdoor attacks require either retraining the classifier with some clean data or modifying the model's architecture. As a result, they are 1) not applicable when clean data is unavailable, 2) less efficient when the model is large, and 3) less stealthy due to architecture changes. In this work, we propose DFBA, a novel retraining-free and data-free backdoor attack without changing the model architecture. Technically, our proposed method modifies a few parameters of a classifier to inject a backdoor. Through theoretical analysis, we verify that our injected backdoor is provably undetectable and unremovable by various state-of-the-art defenses under mild assumptions. Our evaluation on multiple datasets further demonstrates that our injected backdoor: 1) incurs negligible classification loss, 2) achieves 100% attack success rates, and 3) bypasses six existing state-of-the-art defenses. Moreover, our comparison with a state-of-the-art non-data-free backdoor attack shows our attack is more stealthy and effective against various defenses while achieving less classification accuracy loss. The code for our experiment can be found at https://github.com/AAAAAAsuka/DataFree_Backdoor_Attacks

## 1   Introduction

Deep neural networks (DNN) have achieved remarkable success in multiple application domains such as computer vision. To democratize DNN models, especially the powerful but large ones, many machine learning platforms (e.g., ModelZoo [1], TensorFlow Model Garden [2], and Hugging Face [3]) share their pre-trained classifiers to customers with limited resources. For instance, Hugging Face allows any third party to share pre-trained classifiers with the community, which could be downloaded by other users. Despite the benefits brought by those machine learning platforms, existing studies [4, 5, 6] show that this model sharing mechanism is vulnerable to backdoor attacks. In particular, a malicious third party could download a pre-trained classifier from the machine learning platform, inject a backdoor into it, and reshare it with the community via the platform. Backdoor attacks pose severe concerns for the deployment of classifiers downloaded from the machine learning platforms for security and safety-critical applications such as autonomous deriving [7]. We note that the model provider may not share the training data used to train the classifiers when they are trained on private data (e.g., face images).

To thoroughly understand this threat, recent research has proposed a large number of backdoor attacks [4, 5, 6, 8, 9, 10, 11, 12, 13, 14, 15, 16, 17, 18, 19, 20, 21, 22, 23, 24, 25, 26]. At a high level, existing backdoor attacks require either retraining a classifier by accessing some clean data [4, 5, 6, 8, 10, 27] or changing the architecture of a classifier [12, 28]. For instance, Hong et al. [27] proposed a handcrafted backdoor attack, which changes the parameters of a classifier to inject a backdoor. However, they need a set of clean samples that have the same distribution as the training data of the classifier to guide the change of the parameters. Bober-Irizar et al. [28] proposed to inject a backdoor into a classifier by manipulating its architecture. Consequently, their practicality is limited if there is no clean data available, efficiency is restricted if the model is large, or they are less stealthy due to architecture changes. Additionally, none of the existing attacks provide a formal analysis of their attack efficacy against cutting-edge defenses [29, 30]. As a result, they may underestimate the threat caused by backdoor attacks.

**Our contribution:** We propose DFBA, a novel retraining-free and data-free backdoor attack, which injects a backdoor into a pre-trained classifier without changing its architecture. At a high level, DFBA first constructs a *backdoor path* by selecting a single neuron from each layer except the output layer. Then, it modifies the parameters of these neurons such that the backdoor path is activated for a backdoored input but unlikely to be activated for a clean input. As a result, the backdoored classifier predicts backdoored inputs to a target class without affecting the predictions for clean inputs. Specifically, we first optimize a backdoor trigger such that the output of the selected neuron in the first layer is maximized for a backdoored input. Second, we change the parameters of this neuron such that it can only be activated by any input embedded with our optimized trigger but is unlikely to be activated by a clean input. Third, we change the parameters of the middle layer neurons in the backdoor path to gradually amplify the output of the neuron selected in the first layer. Finally, for the output layer's neurons, we change their parameters such that the output of the neurons in the backdoor path has a positive (negative) contribution to the output neuron(s) for the target class (all non-target classes).

We conduct both theoretical and empirical evaluations for DFBA. Theoretically, we prove that backdoors injected by DFBA are undetectable by state-of-the-art detection methods, such as Neural Cleanse [31] and MNTD [30] or irremovable by fine-tuning techniques. Empirically, we evaluate DFBA on various models with different architectures trained from various benchmark datasets. We demonstrate that DFBA can achieve 100% attack success rates across all datasets and models while triggering only less than 3% accuracy loss on clean testing inputs. We also show that DFBA can bypass six state-of-the-art defenses. Moreover, we find that DFBA is more resilient to those defenses than a state-of-the-art non-data-free backdoor attack [27]. Finally, we conduct comprehensive ablation studies to demonstrate DFBA is insensitive to the subtle variations in hyperparameters. To the best of our knowledge, DFBA is the *first* backdoor attack that is retraining-free, data-free, and provides a theoretical guarantee of its attack efficacy against existing defenses.

Our major contributions are summarized as follows:

- We propose DFBA, the first data-free backdoor attack without changing the architecture of a classifier. Our DFBA directly changes the parameters of a classifier to inject a backdoor into it.

- We perform theoretical analysis for DFBA. We show DFBA is provably undetectable or unremovable by multiple state-of-the-art defenses.

- We perform comprehensive evaluations on benchmark datasets to demonstrate the effectiveness and efficiency of DFBA.

- We empirically evaluate DFBA under existing state-of-the-art defenses and find that they are ineffective for DFBA. Our empirical results also show that DFBA is insensitive to hyperparameter choices.

**Roadmap:** We show related work in Section 2, formulate the problem in Section 3, present the technical details of our DFBA in Section 4, show the evaluation results in Section 5, discuss and conclude our DFBA in Section 6.

## 2 Related Work

**Backdoor Attacks.** Existing backdoor attacks either use the whole training set to train a backdoored classifier from scratch [4, 5, 8] or modify the weights or architecture of a pre-trained clean classifier to inject a backdoor [6, 27]. For instance, BadNet [4] constructs a poisoned training set with clean and backdoored data to train a backdoored classifier from scratch. We note that poisoning data based backdoor attacks require an attacker to compromise the training dataset of a model, i.e., inject poisoned data into the training data of the model. Our attack does not have such a constraint. For instance, many machine learning platforms such as Hugging Face allow users to share their models. A malicious attacker could download a pre-trained classifier from Hugging Face, inject a backdoor using our attack, and republish it on Hugging Face to share it with other users. Our attack is directly applicable in this scenario. Moreover, data poisoning based attacks are less stealthy as shown in the previous work [27]. More recent works [6, 12, 19, 21, 27, 28, 32] considered a setup where the attacker has access to a pre-trained clean model rather than the original training dataset. Under this setup, the attacker either manipulates the model's weights with a small set of clean validation data (i.e., parameter modification attacks) or directly vary the model architecture. As discussed in Section 1, those attacks require either retraining with some clean data or modifying the model architecture. In contrast, we propose the first backdoor attack that is entirely retraining-free and data-free without varying the model architecture.

Note that parameter modification attacks share a similar attack mechanism as ours. Among these attacks, some [33, 34, 35] serve a different goal (e.g., fool the model to misclassify certain clean testing inputs) from us. Others [21, 27, 36] still require clean samples to provide guidance for parameter modification. DFBA has the following differences from these methods. First, DFBA does not require data when injecting the backdoor, while these methods still require a few samples. Second, DFBA is provably undetectable and irremovable against various existing defenses (Section B), while existing weight modification attacks do not provide a formal guarantee for its attack efficacy. Finally, as we will show later in Section 5 and Appendix G, compared to the state-of-the-art weight modification attack [27], DFBA incurs less classification accuracy loss on clean testing inputs than [27]. In addition, DFBA requires modifying fewer parameters and is most efficient.

We note that a prior study [26] proposed a "data-free" backdoor attack to deep neural networks. Our method has the following differences with [26]. First, they require the attacker to have a substitution dataset while our method does not have such a requirement (i.e., our method does not require a substitution dataset). Second, they inject the backdoor into a classifier by fine-tuning it. By contrast, our method directly changes the parameters of a classifier to inject the backdoor. Third, they did not provide a formal analysis on the effectiveness of their attack under state-of-the-art defenses.

Recent research has begun to explore data-free backdoor attacks in distributed learning scenarios, particularly in Federated Learning (FL) settings. FakeBA [37] introduced a novel attack where fake clients can inject backdoors into FL systems without real data. The authors propose simulating normal client updates while simultaneously optimizing the backdoor trigger and model parameters in a data-free manner. DarkFeD [38] proposed the first comprehensive data-free backdoor attack scheme. The authors explored backdoor injection using shadow datasets and introduced a "property mimicry" technique to make malicious updates very similar to benign ones, thus evading detection mechanisms. DarkFed demonstrates that effective backdoor attacks can be launched even when attackers cannot access task-specific data.

**Backdoor Detection and Elimination.** Existing defenses against backdoor attacks can be classified into – 1) Training-phase defenses that train a robust classifier from backdoored training data [39, 40, 41, 42]; 2) Deployment-phase defenses that detect and eliminate backdoors from a pre-trained classifier with only *clean data* [31, 43, 44, 45, 46, 47]; 3) Testing-phase defenses [48, 49] that identify the backdoored testing inputs and recover their true prediction result. Training-phase defenses are not applicable to a given classifier that is already backdoored. Testing-phase defenses require accessing to the backdoored inputs (See Section J for more discussion). In this work, we mainly consider the deployment-phase defenses. Existing deployment-phase defenses mainly take three directions: ① detection & removal methods that first reverse-engineer a trigger from a backdoored classifier and then use it to re-train the classifier to unlearn the backdoor [31, 44, 50], ② unlearning methods that fine-tune a classifier with newly collected data to remove the potential backdoors [51, 52, 53, 54], and ③ fine-pruning methods that prune possibly poisoned neurons of the model [51, 55]. As we will

show in Section 5, our attack is empirically resistant to all of these three methods. In addition, under mild assumptions, our attack, with theoretical guarantee, can evade multiple state-of-the-art detection & removal methods and fine-tuning methods (See Section B).

## 3 Problem Formulation

### 3.1 Problem Setup

Consider a pre-trained deep neural network classifier $g$ with $L$ layers, where $\mathbf{W}^{(l)}$ and $\mathbf{b}^{(l)}$ denote its weight and bias at the $l$-th layer. Without loss of generality, we consider ReLU as the activation function for intermediate layers, denoted as $\sigma$, and Softmax as the activation function for the output layer. Given any input that can be flattened into a one-dimensional vector $\mathbf{x} = [x_1, x_2, \cdots, x_d] \in \mathbb{R}^d$, where the value range of each element $x_n$ is $[\alpha_n^l, \alpha_n^u]$, the classifier $g$ maps $\mathbf{x}$ to one of the $C$ classes. For instance, when the pixel value of an image is normalized to the range $[0, 1]$, then we have $\alpha_n^l = 0$ and $\alpha_n^u = 1$. An attacker injects a backdoor into a classifier $g$ such that it predicts any input embedded with an attacker-chosen trigger $(\boldsymbol{\delta}, \mathbf{m})$ as an attacker-chosen target class $y_{tc}$, where $\boldsymbol{\delta}$ and $\mathbf{m}$ respectively represent the pattern and binary mask of the trigger. A backdoored input is represented as follows:

$$\mathbf{x}' = \mathbf{x} \oplus (\boldsymbol{\delta}, \mathbf{m}) = \mathbf{x} \odot (\mathbf{1} - \mathbf{m}) + \boldsymbol{\delta} \odot \mathbf{m}, \tag{1}$$

where $\odot$ represents element-wise multiplication. For simplicity, we denote a classifier injected with the backdoor as $f$. Moreover, given a backdoor trigger $(\boldsymbol{\delta}, \mathbf{m})$, we have $\Gamma(\mathbf{m}) = \{n | m_n = 1, n = 1, 2, \cdots, d\}$, which denotes the set of feature indices where the corresponding value of $\mathbf{m}$ is 1.

### 3.2 Threat Model

**Attacker's goals:** We consider that an attacker aims to implant a backdoor into a target pre-trained model without retraining it or changing its architecture. Meanwhile, we also need to maintain the backdoored model's normal utilities (i.e., performance on clean inputs). Moreover, we require the implanted backdoor to be stealthy such that it cannot be easily detected or removed by existing backdoor detection or elimination techniques.

**Attacker's background knowledge and capability:** Similar to existing attacks [6, 27], we consider the scenarios where the attacker hijacks the ML model supply chain and gains white-box access to a pre-trained model. Differently, *we do not assume that the attacker has any knowledge or access to the training/testing/validation data*. Moreover, we assume that the attacker cannot change the architecture of the pre-trained classifier. In addition, we assume that the attacker does not have access to the training process (e.g., training algorithm and hyperparameters). As discussed above, these assumptions significantly improve the practicability of our proposed attack.

### 3.3 Design Goals

When designing our attack, we aim to achieve the following goals: *utility goal*, *effectiveness goal*, *efficiency goal*, and *stealthy goal*.

**Utility goal:** For the utility goal, we aim to maintain the classification accuracy of the backdoored classifier for clean testing inputs. In other words, the predictions of the backdoored classifier for clean testing inputs should not be affected.

**Effectiveness goal:** We aim to make the backdoored classifier predict the attacker-chosen target label for any testing input embedded with the attacker-chosen backdoor trigger.

**Efficiency goal:** We aim to make the attack that is efficient in crafting a backdoored classifier from a pre-trained clean classifier. We note that an attack that achieves the efficiency goal means it is more practical in the real world.

**Stealthy goal:** The stealthy goal means our attack could bypass existing state-of-the-art defenses. An attack that achieves the stealthy goal means it is less likely to be defended. In this work, we will conduct both theoretical and empirical analysis for our attack under state-of-the-art defenses.

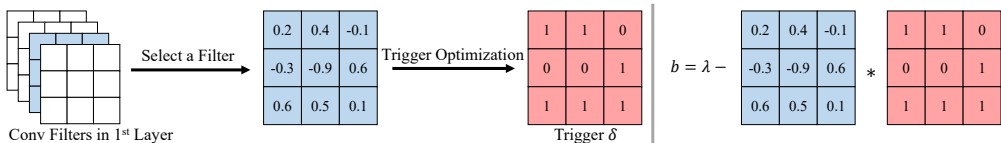

**Figure 1: An example of the backdoor switch and optimized trigger when each pixel of an image is normalized to the range** $[0, 1]$**.**

## 4 DFBA

Since we assume neither model retraining nor architecture modification, the only way of implanting the backdoor is to change the model parameters.[1] Specifically, given a classifier $g$, we aim to manually modify its parameters to craft a backdoored classifier $f$. Our key idea is to create a path (called *backdoor path*) from the input layer to the output layer to inject the backdoor. In particular, our backdoor path satisfies two conditions: 1) it could be activated by any backdoored input such that our backdoor attack is effective, i.e., the backdoored classifier predicts the target class for any backdoored input, and 2) it cannot be activated by a clean input with a high probability to stay stealthy. Our backdoor path involves only a single neuron (e.g. a single filter in CNN) in each layer to reduce the impact of the backdoor on the classification accuracy of the classifier.

The key challenge is how to craft a backdoor path such that it *simultaneously* satisfies the two conditions. To address this challenge, we design a backdoor switch which is a single neuron selected from the first layer of a classifier. We modify the parameters of this neuron such that it will be activated by a backdoored input but is unlikely to be activated by a clean input. Then, we amplify the output of the backdoor switch by changing the parameters of the remaining neurons in the backdoor path. Finally, we change the weights of the output neurons such that the output of the $(L-1)$th-layer neuron in the backdoor path has a positive (or negative) contribution to the output neuron(s) for the target class (or non-target classes) to reach our goal.

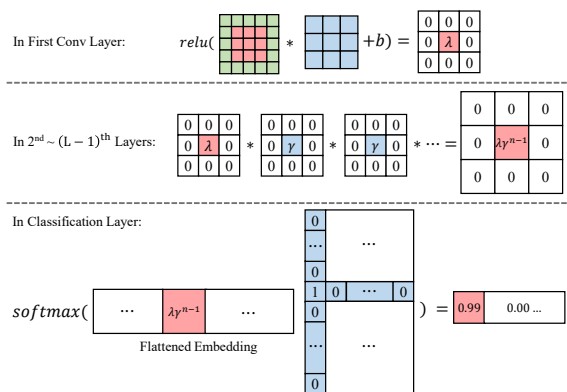

**Figure 2: Visualization of our backdoor path when it is activated by a backdoored input. The backdoored model will predict the target class for the backdoored input.**

### 4.1 Detailed Methodology

#### 4.1.1 Neuron Selection for Backdoor Path

Our goal is to select neurons from a classifier such that they form a path from the first layer to the final output layer. In particular, we select a single neuron from each layer. For the first layer, we randomly select one neuron.[2] As we will see in the next step, neuron selection in this way enables us to change the parameters of the selected neuron such that it has different behaviors for a clean input and a backdoored input. For each middle layer, we select a neuron such that its output depends on the selected neuron in the previous layer. Note that we randomly select one if there exist multiple neurons that satisfy the criteria.

---

[1]Although [27] also does inject the backdoor by changing model parameters, it still requires a few samples to facilitate the parameter changes. While our method does not need any data to inject the backdoor. In addition, we provide a formal guarantee for our attack efficacy.

[2]In fact, we require the neuron's output to depend on the features with the index in $\Gamma(\mathbf{m})$. For a fully connected neural network, it is obvious. For a convolutional neural network, the detail is illustrated in Appendix F.

### 4.1.2 Backdoor Switch

We design a backdoor switch, which is a single neuron (denoted as $s_1$) in the first layer, such that the neuron $s_1$ satisfies two conditions:

**Condition 1:** The switch neuron $s_1$ is activated for a backdoored input $\mathbf{x}'$.

**Condition 2:** The switch neuron $s_1$ is unlikely to be activated for a clean input $\mathbf{x}$.

To achieve the two conditions mentioned above, we need to tackle the following challenges. ❶, given that $x'_n$ can be different for different backdoored inputs for $n \notin \Gamma(\mathbf{m})$. To enable $s_1$ to be activated by any backdoored input, we first need to ensure that the activation of $s_1$ is independent of the value of $x'_n$, where $n \notin \Gamma(\mathbf{m})$. ❷, after we decouple the activation of $s_1$ from $x'_n$, $n \notin \Gamma(\mathbf{m})$, we need to make sure its activation value is only related to the trigger pattern. This is challenging in that the value of $x_n$, where $n \in \Gamma(\mathbf{m})$, can be different for different clean inputs.

**Addressing challenge ❶:** Our key idea is to modify the parameters of the neuron $s_1$ such that its outputs only depend on the features of an input whose indices are in $\Gamma(\mathbf{m})$, i.e., $x_n$ (or $x'_n$) where $n \in \Gamma(\mathbf{m})$. Specifically, we propose to reach this by setting the corresponding weight between $s_1$ and a feature whose index is not in $\Gamma(\mathbf{m})$ to 0. Given an input $\mathbf{x}$, we use $s_1(\mathbf{x})$ to denote the output of the neuron $s_1$. Here, $s_1(\mathbf{x}) = \sigma(\sum w_n x_n + b)$. Given that $w_n = 0$, for $n \notin \Gamma(\mathbf{m})$, we can rewrite $s_1(\mathbf{x}) = \sigma(\sum_{n \in \Gamma(\mathbf{m})} w_n x_n + b)$, which is independent from $x_n$ for $n \notin \Gamma(\mathbf{m})$

**Addressing Challenge ❷:** Our idea is to first optimize the backdoor pattern $\delta_n$ ($n \in \Gamma(\mathbf{m})$) and then modify the remaining parameters of $s_1$ such that 1) $s_1$ is activated for a backdoored input, and 2) $s_1$ is unlikely to be activated when $x_n$ is not close to the optimized $\delta_n$ for $n \in \Gamma(\mathbf{m})$.

Backdoor trigger generation. For a backdoored input, we have $s_1(\mathbf{x}') = \sigma(\sum_{n \in \Gamma(\mathbf{m})} w_n \delta_n + b)$ since $x'_n = \delta_n$ for $n \in \Gamma(\mathbf{m})$. For an arbitrary set of $w_n$ ($n \in \Gamma_{\mathbf{m}}$), we optimize the trigger pattern $\boldsymbol{\delta}$ such that the output of $s_1$ is maximized for a backdoored input, i.e., we aim to solve the following optimization problem:

$$\max_{\boldsymbol{\delta}} \ s_1(\mathbf{x}') = \sigma\left( \sum_{n \in \Gamma(\mathbf{m})} w_n \delta_n + b \right), \text{ s.t. } \alpha_n^l \leq \delta_n \leq \alpha_n^u, \forall n \in \Gamma(\mathbf{m}), \quad (2)$$

where the constraint ensures a backdoored input created by embedding the backdoor trigger $(\boldsymbol{\delta}, \mathbf{m})$ to an arbitrary input is still valid to the classifier, and $[\alpha^l, \alpha^u]$ is the range of feature value $x_n$ (see Section 3.1 for details). Note that although the binary mask $\mathbf{m}$ is chosen by the attacker, we can still derive the analytical solution to the above optimization problem:

$$\delta_n = \begin{cases} \alpha_n^l, & \text{if } w_n \leq 0 \\ \alpha_n^u, & \text{otherwise.} \end{cases} \quad (3)$$

Given the optimized backdoor trigger, we design the following method to modify the bias and weights of $s_1$ to achieve the two conditions.

Activating $s_1$ for $\mathbf{x}'$ by modifying the bias. Recall that our condition 1 aims to make the switch neuron $s_1$ be activated for a backdoored input $\mathbf{x}'$. In particular, given a backdoored input $\mathbf{x}'$ embedded with the trigger $\boldsymbol{\delta}$, the output of the neuron $s_1$ for $\mathbf{x}'$ is as follows: $s_1(\mathbf{x}') = \sigma(\sum_{n \in \Gamma(\mathbf{m})} w_n \delta_n + b)$. To make $s_1$ be activated for a backdoored input $\mathbf{x}'$, we need to ensure $\sum_{n \in \Gamma(\mathbf{m})} w_n \delta_n + b$ to be positive. For simplicity, we denote $\lambda = \sum_{n \in \Gamma(\mathbf{m})} w_n \delta_n + b$. For any $\lambda$, if the bias $b$ of the switch neuron $s_1$ satisfies the above condition, the output of $s_1$ is $\lambda$ for an arbitrary backdoored input. In other words, the switch neuron is activated for a backdoored input, meaning we achieve condition 1. Figure 1 shows an example of our backdoor switch.

Deactivating $s_1$ for $\mathbf{x}$ by modifying the weights. With the above choice of $b$, we calculate the output of the neuron $s_1$ for a clean input $\mathbf{x}$. Formally, we have:

$$s_1(\mathbf{x}) = \sigma\left( \sum_{n \in \Gamma(\mathbf{m})} w_n x_n + \lambda - \sum_{n \in \Gamma(\mathbf{m})} w_n \delta_n \right) = \sigma\left( \sum_{n \in \Gamma(\mathbf{m})} w_n (x_n - \delta_n) + \lambda \right). \quad (4)$$

Our condition 2 aims to make the switch neuron $s_1$ less likely to be activated for a clean input $\mathbf{x}$. Based on Equation 4, we know a clean input $\mathbf{x}$ cannot activate the neuron $s_1$ when $\sum_{n \in \Gamma(\mathbf{m})} w_n (x_n -$

$\delta_n) + \lambda \leq 0$, i.e., $\sum_{n \in \Gamma(\mathbf{m})} w_n(\delta_n - x_n) \geq \lambda$. In other words, when a clean input cannot activate the neuron $s_1$ when it satisfies the following condition: $\sum_{n \in \Gamma(\mathbf{m})} w_n(\delta_n - x_n) \geq \lambda$. By showing this condition is equivalent to $\sum_{n \in \Gamma(\mathbf{m})} |w_n(x_n - \delta_n)| \geq \lambda$, we have the following lemma[3]:

**Lemma 1** *Suppose $\delta_n$ ($n \in \Gamma(\mathbf{m})$) is optimized as in Equation 3. Given an arbitrary clean input $\mathbf{x}$, $\mathbf{x}$ cannot activate $s_1$ if the following condition is satisfied:*

$$\sum_{n \in \Gamma(\mathbf{m})} |w_n(x_n - \delta_n)| \geq \lambda. \tag{5}$$

**Interpretation of $\sum_{n \in \Gamma(\mathbf{m})} |w_n(x_n - \delta_n)| < \lambda$:** We note that $\sum_{n \in \Gamma(\mathbf{m})} |w_n(x_n - \delta_n)| < \lambda$ measures the difference of a clean input and the backdoor trigger for indices in $\Gamma(\mathbf{m})$ (indices where the backdoor trigger is embedded to an input). In particular, for each index in $\Gamma(\mathbf{m})$, $|w_n(x_n - \delta_n)|$ measures the weighted deviation of the feature value of the clean input $\mathbf{x}$ at the dimension $n$ from the corresponding value of the backdoor trigger. Based on the above lemma, a clean input can only activate $s_1$ when $\sum_{n \in \Gamma(\mathbf{m})} |w_n(x_n - \delta_n)| < \lambda$. When $\lambda$ is very small and $|w_n|$ is large, a clean input can only activate the neuron $s_1$ when $x_n$ is very close to $\delta_n$ for $n \in \Gamma_{\mathbf{m}}$, which means that the clean input is very close to its backdoored version. In practice, we find that setting a small $\lambda$ (e.g., 0.1) is enough to ensure a clean input cannot activate $s_1$.

### 4.1.3 Amplifying the Output of the Backdoor Switch

The neuron $s_1$ in the first layer is activated for a backdoored input $\mathbf{x}'$. In the following layers, we can amplify it until the output layer such that the backdoored classifier $f$ outputs the target class $y_{tc}$. Suppose $s_l$ is the selected neuron in the $l$th layer, where $l = 2, 3, \cdots, L - 1$. We can first modify the parameters of $s_l$ such that its output only depends on $s_{l-1}$ and then change the weight between $s_l$ and $s_{l-1}$ to be $\gamma$, where $\gamma$ is a hyperparameter. We call $\gamma$ *amplification factor*. By letting the bias term of $s_l$ to be 0, we have:

$$s_l(\mathbf{x}') = \gamma s_{l-1}(\mathbf{x}'). \tag{6}$$

Note that $s_l(\mathbf{x}) = 0$ when $s_1(\mathbf{x}) = 0$. Finally, we can set the weight between $s_{L-1}$ and the output neuron for the target class $y_{tc}$ to be $\gamma$ but set the weight between $s_l$ and the remaining output neurons to be $-\gamma$. Figure 2 shows an example when the backdoor path is activated by a backdoored input.

### 4.2 Theoretical Analysis

First, we provide the following definitions:

**Pruned classifier:** In our backdoor attack, we select one neuron for each layer in a classifier. Given a pre-trained classifier, we can create a corresponding *pruned classifier* by pruning all the neurons that are selected to form the backdoor path by DFBA. Note that the pruned classifier is clean as it does not have any backdoor.

Based on this definition, we provide the theoretical analysis towards our proposed method in this section. We aim to show that our proposed method can maintain utility on clean data, while cannot be detected by various backdoor model detection methods or disrupted by fine-tuning strategies. Due to space limits, we mainly show the conclusions and guarantees here and leave the details and proof in the Appendix B.

### 4.2.1 Utility Analysis

Our following theorem shows that the backdoored classifier crafted by DFBA has the same output as the pruned classifier for a clean input.

**Theorem 1** *Suppose an input $\mathbf{x}$ cannot activate the backdoor path, i.e., Equation 12 is satisfied for $\mathbf{x}$. Then, the output of the backdoored classifier $g$ for $\mathbf{x}$ is the same as that of the corresponding pruned classifier $h$.*

---
[3]The proof of Lemma 1 can be found in Appendix A.

[**Remark:**] Our above theorem implies that the backdoored classifier has the same classification accuracy as the pruned classifier for clean testing inputs. The pruned classifier is very likely to maintain classification accuracy as we only remove $(L-1)$ neurons for a classifier with $L$ layers. Thus, our DFBA can maintain the classification accuracy of the backdoored classifier for clean inputs.

### 4.2.2 Effectiveness Analysis

In our effectiveness analysis (Section B.2), we show the detection results of query-based defenses [30] and gradient-based defenses [31] for our backdoored classifier are the same as those for the pruned classifier when the backdoor path is not activated, And the following Proposition is given:

**Proposition 1** *Suppose a defense dataset where none of the samples can activate the backdoor path, i.e., Equation 12 is satisfied for each input in the defense dataset. Suppose a defense solely uses the outputs of a classifier for inputs from the defense dataset to detect whether it is backdoored. Then, the same detection result will be obtained for a backdoored classifier and the corresponding pruned classifier.*

**Proposition 2** *Given a classifier, suppose a defense solely leverages the gradient of the output of the classifier with respect to its input to detect whether the classifier is backdoored. If the input cannot activate the backdoor path, i.e., i.e., Equation 12 is satisfied for the input, then the defense produces the same detection results for the backdoored classifier and the pruned classifier.*

[**Remark:**] As the pruned classifier is a clean classifier, our theorem implies that those defenses cannot detect the backdoored classifiers crafted by DFBA.

We also show fine-tuning the backdoored classifier with clean data cannot remove the backdoor:

**Proposition 3** *Suppose we have a dataset $\mathcal{D}_d = \{\mathbf{x}_i, y_i\}_{i=1}^{N}$, where each sample $\mathbf{x}_i$ cannot activate the backdoor path, i.e., Equation 12 is satisfied for each $\mathbf{x}_i$. Then, the parameters of the neurons that form the backdoor path will not be affected if the backdoored classifier is fine-tuned using the dataset $\mathcal{D}_d$.*

All the complete proof and analysis process can be found in the Appendix B

## 5   Evaluation

We perform comprehensive experiments to evaluate our DFBA. In particular, we consider 1) multiple benchmark datasets, 2) different models, 3) comparisons with state-of-the-art baselines, 4) evaluation of our DFBA under 6 defenses (i.e., Neural Cleanse [31], Fine-tuning [51], and Fine-pruning [51], MNTD [30], I-BAU [53], Lipschitz pruning [55]), and 5) ablation studies on all hyperparameters. Our experimental results show that 1) our DFBA can achieve high attack success rates while maintaining the classification accuracy on all benchmark datasets for different models, 2) our DFBA outperforms a non-data-free baseline, 3) our DFBA can bypass all 6 defenses, 4) our DFBA is insensitive to hyperparameters, i.e., our DFBA is consistently effective for different hyperparameters.

### 5.1   Experimental Setup

**Models:**  We consider a fully connected neural network (FCN) and a convolutional neural network (CNN) for MNIST and Fashion-MNIST. The architecture can be found in Table VI in the Appendix. By default, we use CNN on those two datasets. We consider VGG16 [56] and ResNet-18 [57] for CIFAR10 and GTSRB, respectively. We use ResNet-50 and ResNet-101 for ImageNet.

**Evaluation metrics:**  Following previous work on backdoor attacks [4, 6], we use *clean accuracy (CA)*, *backdoored accuracy (BA)*, and *attack success rate (ASR)* as evaluation metrics. For a backdoor attack, it achieves the utility goal if the backdoored accuracy is close to the clean accuracy. A high ASR means the backdoor attack achieves the effectiveness goal. For the efficiency goal, we use the computation time to measure it. Additionally, when we evaluate defenses, we further use *ACC* as an evaluation metric, which is the classification accuracy on clean testing inputs of the classifier obtained after the defense.

**Compared methods:** We compare our DFBA with the state-of-the-art handcrafted backdoor attack [27], which changes the parameters of a pre-trained classifier to inject a backdoor. We note that Hong et al. [27] showed that their attack is more robust against defenses compared with traditional data poisoning based backdoor attacks [4]. So, we only compare with Hong et al. [27].

### 5.2 Experimental Results

**Our DFBA maintains classification accuracy:** Table 1 compares the CA and BA of our method. The results show that BA is comparable to CA. In particular, the difference between BA and CA is less than 3% for different datasets and models, i.e., our attack maintains the classification accuracy of a machine learning classifier. The reasons are as follows: 1) our backdoor path only consists of a single neuron in each layer of a classifier, and 2) we find that (almost) no clean testing inputs can activate the backdoor path on all datasets and models as shown in Table 5. We note that the classification accuracy loss on ImageNet is slightly larger than those on other datasets. We suspect the reason is that ImageNet is more complex and thus randomly selection neurons are more likely to impact classification accuracy. As part of future work, we will explore methods to further improve classification accuracy, e.g., designing new data-free methods to select neurons from a classifier.

**Our DFBA achieves high ASRs:** Table 1 shows the ASRs of our attack for different datasets and models. Our experimental results show that our attack can achieve high ASRs. For instance, the ASRs are 100% on all datasets for all different models. The reason is that all backdoored testing inputs can activate our backdoor path as shown in Table 5. Once our backdoor path is activated for a backdoored testing input, the backdoored classifier crafted by our DFBA would predict the target class for it. Our experimental results demonstrate the effectiveness of our DFBA.

**Our DFBA is efficient:** Our attack directly changes the parameters of a classifier to inject a backdoor and thus is very efficient. We evaluate the computation cost of our DFBA. For instance, without using any GPUs, it takes less than 1s to craft a backdoored classifier from a pre-trained classifier on all datasets and models. For

Table 1: Our attack is effective while maintaining utility.

| Dataset | Model | CA (%) | BA (%) | ASR (%) |
|---|---|---|---|---|
| MNIST | FCN | 96.49 | 95.51 | 100.00 |
| | CNN | 99.03 | 99.01 | 100.00 |
| Fashion-MNIST | FCN | 81.30 | 81.01 | 100.00 |
| | CNN | 90.09 | 89.55 | 100.00 |
| CIFAR10 | VGG16 | 92.22 | 91.67 | 100.00 |
| | ResNet-18 | 92.16 | 91.33 | 100.00 |
| GTSRB | VGG16 | 95.83 | 95.76 | 100.00 |
| | ResNet-18 | 96.74 | 96.70 | 100.00 |
| ImageNet | ResNet-50 | 76.13 | 73.51 | 100.00 |
| | ResNet-101 | 77.38 | 74.70 | 100.00 |

example, On an NVIDIA RTX A100 GPU, DFBA injects backdoors in 0.0654 seconds for ResNet-18 model trained on CIFAR10, and 0.0733 seconds for ResNet-101 trained on ImageNet. In contrast, similar methods, such as Lv et al. [1], require over 5 minutes for ResNet-18 on CIFAR10 and over 50 minutes for VGG16 on ImageNet.

**Our DFBA outperforms existing non-data-free attacks:** We compare with state-of-the-art non-data-free backdoor attacks [27]. In our comparison, we use the same setting as Hong et al. [27]. We randomly sample 10,000 images from the training dataset to inject the backdoor for Hong et al. [27]. Table 2 shows the comparison results on MNIST. We have two observations. First, our DFBA incurs small classification loss than Hong et al. [27]. Second, our DFBA achieves higher ASR than Hong et al. [27]. Our experimental results demonstrate that our DFBA can achieve better performance than existing state-of-the-art non-data-free backdoor attack [27].

**Our DFBA is effective under state-of-the-art defenses:** Recall that existing defenses can be categorized into three types (See Section 2 for details): backdoor detection, unlearning methods, and pruning methods. For each type, we select two methods, which are respectively the most representative and the state-of-the-art methods. We compare DFBA with Hong et al. [27] for three representative methods (i.e., Neural Cleanse [31], Fine-tuning [51], and Fine-pruning [51]) on MNIST. We adopt the same model architecture as used by Hong et al. [27] in our comparison. We

evaluate three additional state-of-the-art defenses for DFBA (i.e., MNTD [30], I-BAU [53], Lipschitz pruning [55]). All these experiments results and analysis can be find in Appendix D, In summary, our DFBA can consistently bypass those three defenses.

# 6 Conclusion

In this work, we design DFBA, a novel retraining-free and data-free backdoor attack without changing the architecture of a pre-trained classifier. Theoretically, we prove that DFBA can evade multiple state-of-the-art defenses under mild assumptions. Our evaluation on various datasets shows that DFBA is more effective than existing attacks in attack efficacy and utility maintenance. Moreover, we also evaluate the effectiveness of DFBA under multiple state-of-the-art defenses. Our re-

Table 2: Comparing DFBA with state-of-the-art non-data-free backdoor attack [27].

| Method | CA (%) | BA (%) | ASR (%) |
|---|---|---|---|
| Hong et al. [27] | 96.49 | 95.29 | 94.59 |
| DFBA | 96.49 | **95.51** | **100.00** |

sults show those defenses cannot defend against our attacks. Our ablation studies further demonstrate that DFBA is insensitive to hyperparameter changes. Promising future work includes 1) extending our attack to other domains such as natural language processing (NLP), 2) designing different types of triggers for our backdoor attacks, and 3) generalizing our attack to transformer architecture.

# 7 Acknowledgments

This research is supported in part by ARL funding W911NF-23-2-0137, Singapore National Research Foundation funding 053424, DARPA funding 112774-19499.

This material is in part based upon work supported by the National Science Foundation under grant no. 2229876 and is supported in part by funds provided by the National Science Foundation, by the Department of Homeland Security, and by IBM.

Any opinions, findings, and conclusions or recommendations expressed in this material are those of the author(s) and do not necessarily reflect the views of the National Science Foundation or its federal agency and industry partners.

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

# A Proof of Lemma 1

[Proof of Lemma 1] A clean input $\mathbf{x}$ cannot activate the neuron $s_1$ when $\sum_{n \in \Gamma(\mathbf{m})} w_n(x_n - \delta_n) + \lambda \leq 0$, i.e., $\sum_{n \in \Gamma(\mathbf{m})} w_n(\delta_n - x_n) \geq \lambda$. We prove this condition is equivalent to $\sum_{n \in \Gamma(\mathbf{m})} |w_n(x_n - \delta_n)| \geq \lambda$. Suppose $w_n \leq 0$, then we know $\delta_n = \alpha_n^l$ based on Equation 3. Since $x_n \in [\alpha_n^l, \alpha_n^u]$, we know $x_n \geq \delta_n$. Therefore, we have $w_n(\delta_n - x_n) \geq 0$, i.e., $w_n(\delta_n - x_n) = |w_n(\delta_n - x_n)|$. Similarly, we can show that $w_n(\delta_n - x_n) = |w_n(\delta_n - x_n)|$ when $w_n > 0$. Therefore, we have $\sum_{n \in \Gamma(\mathbf{m})} w_n(\delta_n - x_n) = \sum_{n \in \Gamma(\mathbf{m})} |w_n(x_n - \delta_n)|$, i.e., the condition that a clean input cannot activate $s_1$ is as follows:

$$\sum_{n \in \Gamma(\mathbf{m})} |w_n(x_n - \delta_n)| \geq \lambda, \tag{7}$$

where $\lambda$ is a hyperparameter.

# B Theoretical Analysis

From Lemma 1, we know that a clean input $\mathbf{x}$ cannot activate the backdoor path if and only if the following equation is satisfied:

$$\sum_{n \in \Gamma(\mathbf{m})} |w_n(x_n - \delta_n)| \geq \lambda. \tag{8}$$

In other words, $\mathbf{x}$ can only activate the backdoor path if we have $\sum_{n \in \Gamma(\mathbf{m})} |w_n(x_n - \delta_n)| < \lambda$. Suppose that the input $\mathbf{x}$ is sampled from a certain distribution. We use $p$ to denote the probability that an input $\mathbf{x}$ can activate our injected backdoor in a classifier. Then, we have:

$$p = \Pr\left( \sum_{n \in \Gamma(\mathbf{m})} |w_n(x_n - \delta_n)| < \lambda \right). \tag{9}$$

This probability is very small when $\lambda$ is small and $w_n$ is large. We have the following example when each entry of $\mathbf{x}$ follows uniform distribution.

**Example 1** *Suppose $x_n$ ($n = 1, 2, \cdots, d$) follows a uniform distribution over $[0, 1]$. Moreover, we assume $x_n$ to be i.i.d.. When $|w_n| \geq \alpha$ for $n \in \Gamma(\mathbf{m})$, we have $p \leq \frac{(2\lambda)^e}{\alpha^e e!}$, where $e$ is the number of elements in $\Gamma(\mathbf{m})$.*

When $|w_n| \geq \alpha$, we have:

$$p = \Pr\left( \sum_{n \in \Gamma(\mathbf{m})} |w_n(x_n - \delta_n)| < \lambda \right) \tag{10}$$

$$\leq \Pr\left( \sum_{n \in \Gamma(\mathbf{m})} |x_n - \delta_n| < \lambda/\alpha \right). \tag{11}$$

Moreover, since $x_n$ follows a uniform distribution between 0 and 1, the probability $p$ is no larger than the volume of an $\ell_1$-ball with radius $\lambda/\alpha$ in the space $\mathbb{R}^e$, where $e$ is the number of elements in $\Gamma(\mathbf{m})$. The volume can be computed as $\frac{(2\lambda)^e}{\alpha^e e!}$. Thus, we have $p \leq \frac{(2\lambda)^e}{\alpha^e e!}$. We have the following remarks from our above example:

- As a concrete example, we have $p \leq 3.13 \times 10^{-9}$ when $\lambda = 1$, $\alpha = 1$, and $e = 16$ for a $4 \times 4$ trigger.

- In practice, $\mathbf{x}$ may follow a different distribution. We empirically find that almost all testing examples cannot activate the backdoor path when $\lambda$ is small (e.g., 0.1) on various benchmark datasets, indicating that it is hard in general for regular data to activate the backdoor path. As we will show, this enables us to perform theoretical analysis on the utility and effectiveness of the backdoored classifier by DFBA.

## B.1 Utility Analysis

Given an input $\mathbf{x} = [x_1, x_2, \cdots, x_d]$ and a backdoored classifier $g$ crafted by our attack. Based on Lemma 1, we know the input $\mathbf{x}$ cannot activate the backdoor path if the following condition is satisfied:

$$\sum_{n \in \Gamma(\mathbf{m})} |w_n(x_n - \delta_n)| \geq \lambda, \tag{12}$$

where $x_n$ is the feature value of $\mathbf{x}$ at the $n$th dimension, $w_n$ is the weight between the first neuron in the backdoor path of the backdoored classifier $g$ and $x_n$, $\Gamma(\mathbf{m})$ is a set of indices of the location of the backdoor trigger, and $\delta_n$ ($n \in \Gamma(\mathbf{m})$) is the value of the backdoor pattern. The above equation means a clean input $\mathbf{x}$ cannot activate the backdoor path when the weighted sum of its deviation from the backdoor trigger is no smaller than $\lambda$ (a hyper-parameter).

If $\mathbf{x}$ cannot activate the backdoor path, the outputs of the neurons in the backdoor path are 0. Thus, the output of the backdoored classifier does not change if those neurons are pruned. As a result, the prediction of the backdoored classifier for $\mathbf{x}$ is the same as that of the pruned classifier.

## B.2 Attack Efficacy Analysis

We will theoretically analyze the performance of our DFBA under various backdoor defenses.

### B.2.1 Undetectable Analysis

We consider two types of defenses: query-based defenses [30] and gradient-based defenses [31].

[Proof of Proposition 1] Based on Theorem 1, the output of the backdoored classifier is the same as the pruned classifier if an input cannot activate the backdoor path. Thus, the output for any input from the defense dataset will be the same for the two classifiers, which leads to the same detection result.

[Proof of Proposition 2] When inputs cannot activate backdoor path, gradients of outputs of the backdoored classifier and pruned classifier with respect to their inputs are the same. Thus, detection results are same.

**Pruning-based defenses [51, 55]:** We note that a defender can prune the neurons whose outputs on clean data are small or Lipschitz constant is large to mitigate our attack [51, 55]. As we will empirically show in Section 5, our DFBA can be adapted to evade those defenses. Moreover, we empirically find that our adaptive attack designed for pruning-based defenses can also evade other defenses such as Neural Cleanse and MNTD (see Section 5 for details). Therefore, we can use our adaptive attack in practice if we don't have any information on the defense.

### B.2.2 Unremovable Analysis

The goal of backdoor removal is to remove the backdoor in a classifier. For instance, fine-tuning is widely used to remove the backdoor in a classifier. Suppose we have a dataset $\mathcal{D}_d = \{\mathbf{x}^i, y^i\}_{i=1}^N$. Given a classifier $f$, fine-tuning aims to train it such that it has high classification accuracy on $\mathcal{D}_d$. Formally, we have the following optimization problem:

$$\min_{f'} \frac{1}{|\mathcal{D}_d|} \sum_{(\mathbf{x},y) \in \mathcal{D}_d} \ell(f'(\mathbf{x}), y),$$

where $\ell$ is the loss function, e.g., cross-entropy loss, and $f'$ is initialized with $f$. We can use SGD to solve the optimization problem. However, fine-tuning is *ineffective* against our DFBA. Formally, we have:

[Proof of Proposition 3] Given that 1) each training input cannot activate the backdoor path, and 2) the output of the neurons in the backdoor path is independent of the neurons that are not in the backdoor path, the gradient of loss function with respect to parameters of the neurons in the backdoor path is 0. Thus, the parameters of those neurons do not change.

**Table 3: The neural network architectures for MNIST and FashionMNIST.**

(a) FCN

| Layer Type | Layer Parameters |
|---|---|
| Input 784 | |
| Linear | output shape: 32 |
| Activation | $ReLU$ |
| Output 10 | |

(b) CNN

| Layer Type | Layer Parameters |
|---|---|
| Input $28 \times 28$ | |
| Convolution | $16 \times 5 \times 5$, |
| | strides=$(1, 1)$, padding=None |
| Activation | $ReLU$ |
| Convolution | $32 \times 5 \times 5$, |
| | strides=$(1, 1)$, padding=None |
| Activation | $ReLU$ |
| MaxPool2D | kernel size=$(2, 2)$ |
| Flatten | |
| Linear | output shape: 1024 |
| Activation | $ReLU$ |
| Output 10 | |

## C   More Details of Experiments

**Datasets:**  We consider the following benchmark datasets: MNIST, Fashion-MNIST, CIFAR10, GTSRB, and ImageNet.

- **MNIST:** MNIST dataset is used for digit classification. In particular, the dataset contains 60,000 training images and 10,000 testing images, where the size of each image is $28 \times 28$. Moreover, each image belongs to one of the 10 classes.

- **Fashion-MNIST:** Fashion-MNIST is a dataset of Zalando's article images. Specifically, the dataset contains 60,000 training images and 10,000 testing images. Each image is a $28 \times 28$ grayscale image and has a label from 10 classes.

- **CIFAR10:** This dataset is used for object recognition. The dataset consists of 60,000 $32 \times 32 \times 3$ colour images, each of which belongs to one of the 10 classes. The dataset is divided into 50,000 training images and 10,000 testing images.

- **GTSRB:** This dataset is used for traffic sign recognition. The dataset contains 26,640 training images and 12,630 testing images, where each image belongs to one of 43 classes. The size of each image is $32 \times 32 \times 3$.

- **ImageNet:** The ImageNet dataset is used for object recognition. There are 1,281,167 training images and 50,000 testing images in the dataset, where each image has a label from 1,000 classes. The size of each image is $224 \times 224 \times 3$.

Table 4 summarizes the statistics of those datasets. Unless otherwise mentioned, we use MNIST dataset in our evaluation.

**Parameter settings:**  We conducted all experiments on an NVIDIA A100 GPU, and the random seed for all experiments was set to 0. Our attack has the following parameters: threshold $\lambda$, amplification factor $\gamma$, and trigger size. Unless otherwise mentioned, we adopt the following default parameters: we set $\lambda = 0.1$. Moreover, we set $\gamma$ to satisfy $\lambda \gamma^{L-1} = 100$, where $L$ is the total number of layers of a neural network. In Figure 7, we conduct an ablation study on $\lambda$ and $\gamma$. We find that $\lambda$ and $\gamma$ could influence the utility of a classifier and attack effectiveness. When $\lambda$ is small, our method would not influence utility. When $\gamma$ is large, our attack could consistently achieve a high attack success rate. Thus, in practice, we could set a small $\lambda$ and a large $\gamma$.

We set the size of the backdoor trigger (in the bottom right corner) to $4 \times 4$ and the target class to 0 for all datasets. In our ablation studies, we will study their impact on our attack. In particular, we set

**Table 4: Dataset statistics.**

| Dataset | #Training images | #Testing images | #Classes |
|---|---|---|---|
| MNIST | 60,000 | 10,000 | 10 |
| Fashion-MNIST | 60,000 | 10,000 | 10 |
| CIFAR10 | 50,000 | 10,000 | 10 |
| GTSRB | 26,640 | 12,630 | 43 |
| ImageNet | 1,281,167 | 50,000 | 1,000 |

**Table 5: Number of clean testing inputs and backdoored testing inputs that can activate our backdoor path.**

| Model | Dataset | Clean Testing Input | Backdoored Testing Input |
|---|---|---|---|
| FCN | MNIST | 0 / 10,000 | 10,000 / 10,000 |
| | Fashion-MNIST | 0 / 10,000 | 10,000 / 10,000 |
| CNN | MNIST | 0 / 10,000 | 10,000 / 10,000 |
| | Fashion-MNIST | 1 / 10,000 | 10,000 / 10,000 |
| VGG16 | CIFAR10 | 0 / 10,000 | 10,000 / 10,000 |
| | GTSRB | 0 / 12,630 | 12,630 / 12,630 |
| ResNet-18 | CIFAR10 | 0 / 10,000 | 10,000 / 10,000 |
| | GTSRB | 0 / 12,630 | 12,630 / 12,630 |
| ResNet-50 | ImageNet | 0 / 50,000 | 50,000 / 50,000 |
| ResNet-101 | ImageNet | 0 / 50,000 | 50,000 / 50,000 |

all other parameters to their default values when studying the impact of one parameter. Note that our trigger pattern is calculated via solving the optimization problem in Equation 2, whose solution can be found in Equation 3. Figure 8 (in Appendix K) visualizes the trigger pattern.

## D   Effectiveness of DFBA Under State-of-the-art Defenses

**Our DFBA cannot be detected by Neural Cleanse [31]:**  Neural Cleanse (NC) leverages a clean dataset to reverse engineer backdoor triggers and use them to detect whether a classifier is backdoored. In our experiments, we use the training dataset to reverse engineer triggers. We adopt the publicly available code [29] in our implementation. We train 5 clean classifiers and then respectively craft 5 backdoored classifiers using DFBA and Hong et al. [27]. We report the detection rate which is the fraction of backdoored classifiers that are correctly identified by NC for each method. The detection rate of NC for DFBA is 0. In contrast, NC can achieve 100% detection rate for Hong et al. [27] based on the results in Figure 9 in Hong et al. [27] (our setting is the same as Hong et al. [27]). Therefore, our DFBA is more stealthy than Hong et al. [27] under NC. The reason why NeuralCleanse does not work is as follows. NeuralCleanse uses a validation dataset to reverse engineer a trigger such that a classifier is very likely to predict the target class when the trigger is added to inputs in the validation dataset. However, our backdoor path is very hard to be activated by non-backdoored inputs (as shown in Table 5). In other words, our backdoor path is not activated when NeuralCleanse reverse engineers the trigger, which makes NeuralCleanse ineffective.

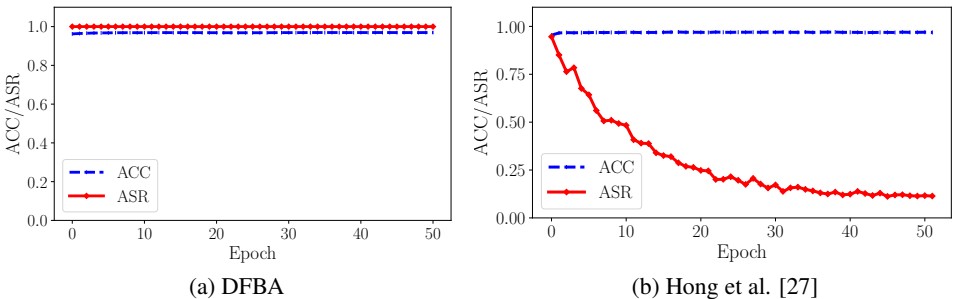

(a) DFBA

(b) Hong et al. [27]

Figure 3: Comparing DFBA with Hong et al. [27] under fine-tuning.

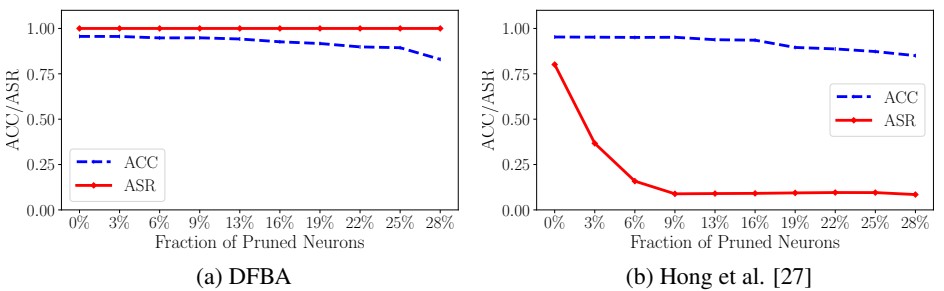

(a) DFBA

(b) Hong et al. [27]

Figure 4: Comparing DFBA with Hong et al. [27] under pruning [51].

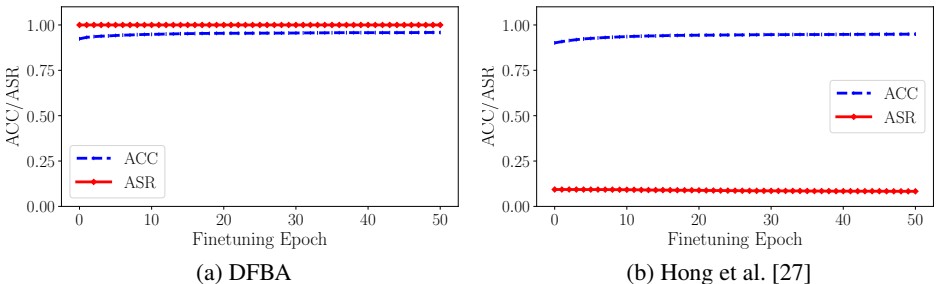

(a) DFBA

(b) Hong et al. [27]

Figure 5: Comparing our DFBA with Hong et al. [27] under fine-tuning after pruning neurons on MNIST.

**Our DFBA is resilient to fine-tuning:** Given a backdoored model, a defender can use clean data to fine-tune it to remove the backdoor. To consider a strong defense, we use the entire training dataset of MNIST to fine-tune the backdoored classifier, where the learning rate is 0.01. Figure 3 shows the experimental results of Hong et al. [27] and DFBA. We find that the ASR of DFBA remains high when fine-tuning the backdoored classifier for different epochs. In contrast, the ASR of Hong et al. [27] decreases as the number of fine-tuning epochs increases.

**Our DFBA is resilient to fine-pruning:** Liu et al. [51] proposed to prune neurons whose outputs are small on a clean dataset in a middle layer of a classifier to remove the backdoor. Our DFBA can be adapted to evade this defense. Suppose we have a clean validation dataset, Liu et al. [51] proposed to remove neurons whose outputs are small in a certain middle layer (e.g., the last fully connected layer in a fully connected neural network). Our DFBA can be adapted to evade this attack. Our idea is to let both clean and backdoored inputs activate our backdoor path. As we optimize the backdoor trigger, the outputs of neurons on backdoored inputs are much larger than those on clean inputs. Thus,

**Table 6: Our attack is effective under I-BAU [53].**

| Model | Model | ACC | ASR (%) |
|---|---|---|---|
| FCN | Before Defense | 95.51 | 100.00 |
| | After Defense | 95.58 | 100.00 |
| CNN | Before Defense | 99.01 | 100.00 |
| | After Defense | 93.05 | 100.00 |

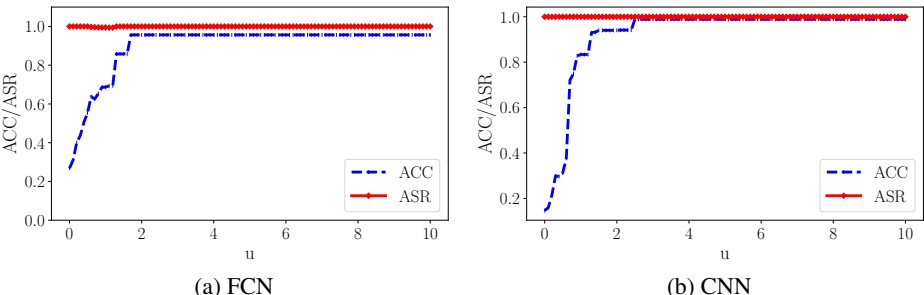

(a) FCN        (b) CNN

**Figure 6: The ACC and ASR of our attack under Lipchitz Pruning on MNIST.**

our adapted backdoor attack is effective while maintaining classification accuracy on clean inputs. In particular, we randomly sample from a zero-mean Gaussian distribution $\mathcal{N}(0, \sigma^2)$ as parameters that are related to features whose indices in $\Gamma(\mathbf{m})$ for the selected neuron in the first layer, where $\sigma$ is the standard deviation of Gaussian noise. Moreover, we don't change the bias of the selected neuron in the first layer such that both clean inputs and backdoored inputs can activate the backdoor path. In our experiments, we set $\sigma = 4,000$ and $\gamma = 1$. Note that we set $\gamma = 1$ because the output of the neuron selected from the first layer is already very large for a backdoored input.

We prune neurons whose outputs are small on the training dataset. Figure 4 shows results for DFBA and Hong et al. [27]. We find that DFBA can consistently achieve high ASR when we prune different fractions of neurons. In contrast, the ASR of Hong et al. [27] decreases as more neurons are pruned. We further fine-tune the pruned model (we prune neurons until the ACC drop is up to 5%) using the training dataset. Figure 5 shows the results. We find that DFBA can still achieve high ASR after fine-tuning.

**MNTD [30] cannot detect DFBA:** MNTD trains a meta classifier to predict whether a classifier is backdoored or not. Roughly speaking, the idea is to train a set of clean models and backdoored models. Specifically, given a set of inputs (called *query set*) and a model, MNTD uses the output of the model on the query set as its feature. Then, a meta-classifier is trained to distinguish clean models and backdoored models based on their features. Note that they also optimize the query set to boost the performance.

We evaluate the performance of MNTD for DFBA on MNIST. We use the publicly available code of MNTD in our experiments[4]. We respectively train 5 clean classifiers using different seeds and then craft 5 backdoored classifiers using DFBA for FCN and CNN. We use the detection rate as the evaluation metric, which measures the fraction of backdoored classifiers that are correctly identified by MNTD. We find the detection rate of MNTD is 0 for both FCN and CNN, i.e., MNTD is ineffective for DFBA. Our empirical results are consistent with our theorem (Proposition 1).

**I-BAU [53] cannot remove DFBA's backdoor:** Zeng et al. [53] proposed I-BAU, which aims to unlearn the backdoor in a classifier. I-BAU formulates the backdoor unlearn as a minimax optimization problem. In the inner optimization problem, I-BAU aims to find a trigger such that the classifier has a high classification loss when the trigger is added to clean inputs. In the outer

---

[4]https://github.com/AI-secure/Meta-Nerual-Trojan-Detection

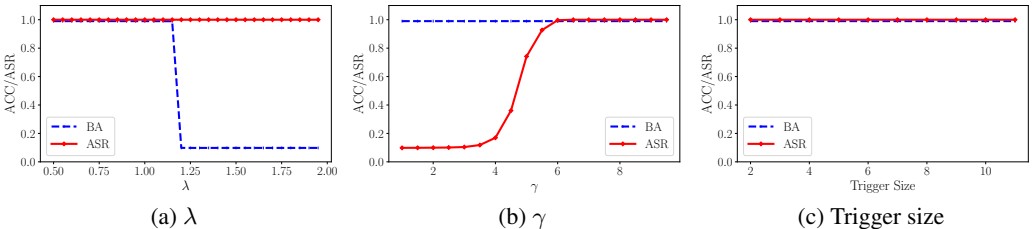

(a) $\lambda$            (b) $\gamma$            (c) Trigger size

**Figure 7: Impact of $\lambda$, $\gamma$, and trigger size on DFBA.**

optimization problem, I-BAU aims to re-train the classifier such that it has high classification accuracy on clean inputs added with the optimized trigger.

We apply I-BAU to unlearn the backdoor injected by DFBA on MNIST. We use the publicly available code in our implementation[5]. Table 6 shows our experimental results. We find that the ASR is still very high after applying I-BAU to unlearn the backdoor in the classifier injected by DFBA. Our results demonstrate that I-BAU cannot effectively remove the backdoor.

**DFBA can be adapted to evade Lipschitz Pruning [55]:** Zheng et al. [55] proposed to leverage Lipschitz constant to prune neurons in a classifier to remove the backdoor. In particular, for the $k$th convolutional layer, Zheng et al. [55] proposed to compute a Lipschitz constant for each convolution filter. Then, Zheng et al. [55] compute the mean (denoted as $\mu_k$) and standard deviation (denoted as $\sigma_k$) of those Lipschitz constants. The convolution filters whose Lipschitz constants are larger than $\mu_k + u\sigma_k$ are pruned, where $u$ is a hyperparameter. The method can be extended to a fully connected layer by computing a Lipschitz constant for each neuron.

Our DFBA can be adapted to evade [55]. In particular, we set $\gamma$ to be a small value for the neurons selected in the middle layers such that its Lipschitz constant is smaller than $\mu_k$. To ensure the effectiveness of backdoor attacks, our idea is to change the parameter of the neurons in the output layer. In particular, we can set the weight between $s_{L-1}$ and the output neuron for the target class $y_{tc}$ to be a larger number but set the weight between $s_l$ and the remaining output neurons to be a small number. Note that the neurons in the output layer are not pruned in [55]. Figure 6 shows our experimental results. We find that our DFBA can consistently achieve high ASR for different $u$, which demonstrates the effectiveness of our backdoor attacks. We note that the ACC is low for small $u$ because more neurons are pruned by Lipschitz Pruning [55] when $u$ is smaller.

**Effectiveness of our adaptive attacks tailored to pruning-based defenses for other defenses:** Our attack requires the attacker to know the defense information to have a formal guarantee of the attack efficacy under those defenses. When the attacker does not have such information, the attacker can use our adaptive attack designed for pruning-based defenses in practice. We performed evaluations under our default setting to validate this. We find that our adaptive attack designed for fine-pruning can also evade Neural Cleanse, fine-tuning, MNTD, I-BAU, and Lipschitz pruning. In particular, the detection rate of both Neural Cleanse and MNTD for backdoored classifiers crafted by DFBA is 0% (we apply the detection on five backdoored classifiers and report the detection accuracy as the fraction of backdoored classifiers that are detected by each method), which means they cannot detect backdoored classifiers. The attack success rate (ASR) is still 100% after we fine-tune the backdoored classifier for 50 epochs (or use I-BAU to unlearn the backdoor or use Lipschitz pruning to prune neurons to remove the backdoor). Our results demonstrate that our adaptive attack can be used when the information on the defense is unavailable.

## E Ablation Studies

We perform ablation studies to study the impact of hyperparameters of our DFBA. In particular, our DFBA has the following hyperparameters: threshold $\lambda$, amplification factor $\gamma$, and trigger size. When

---

[5]https://github.com/YiZeng623/I-BAU

we study the impact of each hyperparameter, we set the remaining hyperparameters to their default values.

**Impact of $\lambda$:** Figure 7a shows the impact of $\lambda$ on MNIST. We have the following observations. First, our attack consistently achieves high ASR. The reason is that the backdoor path crafted by DFBA is always activated for backdoored inputs when $\lambda > 0$. Second, DFBA achieves high BA when $\lambda$ is very small, i.e., DFBA can maintain the classification accuracy of the backdoored classifier for clean testing inputs when $\lambda$ is small. Third, BA decreases when $\lambda$ is larger than a threshold. This is because the backdoored path can also be activated by clean inputs when $\lambda$ is large. As a result, those clean inputs are predicted as the target class which results in the classification loss. Thus, we can set $\lambda$ to be a small value in practice, e.g., 0.1.

**Impact of $\gamma$:** Figure 7b shows the impact of $\gamma$ on MNIST. The ASR of DFBA first increases as $\gamma$ increases and then becomes stable. The reason is that the output of neurons in the backdoor path is larger for a backdoored input when $\gamma$ is larger. As a result, the backdoored input is more likely to be predicted as the target class. Thus, we can set $\gamma$ to be a large value in practice.

**Impact of trigger size:** Figure 7c shows the impact of trigger sizes on MNIST. We find that our backdoor attack can consistently achieve high ASR and BA for backdoor triggers with different sizes. For instance, our attack could still achieve a 100% ASR when the size of the trigger is $2 \times 2$.

**Impact of trigger location:** We note that our attack is also effective even if the trigger position changes for convolutional neural networks. The reason is that a convolutional filter is applied in different locations of an image to perform convolution operation. Thus, the output of the convolution filter would be large when the trigger is present and thus activate our back path, making our attack successful. We also validate this by experiments. For instance, we find that our attack could still achieve a 100% ASR when we change the location of the trigger under the default setting.

# F    Neuron Selection for a CNN

For a convolutional neural network, a convolution filter generates a channel for an input. In particular, each value in the channel represents the output of one neuron, where all neurons whose outputs are in the same channel share the same parameters. We randomly select one neuron whose output value depends on the features with indices in $\Gamma(\mathbf{m})$. We note that, as neurons in the same channel share the parameters, they would be affected if we change the parameters of one neuron. We consider this when we design our DFBA. As a result, our DFBA can maintain the classification accuracy of the classifier for normal testing inputs as shown in our experimental results.

# G    Comparing with Hong et al. [27] on CIFAR10 Dataset

We also compare our attack with Hong et al. on CIFAR10 dataset, where the classifier is CNN. We compare DFBA with Hong et al. for fine-tuning and fine-pruning. Our comparison results are as follows. After fine-tuning, the ASRs of our DFBA and Hong et al. are 100% and 88%, respectively. After fine-pruning, the ASRs of our DFBA and Hong et al. are 100% and 84%, respectively. Our results demonstrate that our attack is more effective than Hong et al.. Our observations on CIFAR10 are consistent with those on MNIST.

# H    Evaluation of Neural Cleanse, MNTD, I-BAU, and Lipschitz pruning against Our Attack on CIFAR10

We also evaluate other defenses on CIFAR10, including Neural Cleanse, MNTD, I-BAU, and Lipschitz pruning against our attack. The detection accuracy (we apply the detection on five backdoored classifiers and report the detection accuracy as the fraction of backdoored classifiers that are detected by each method) of Neural Cleanse and MNTD is 0% for our DFBA. Our DFBA can still achieve a 100% ASR after we apply Lipschitz pruning to the backdoored classifier. We find that I-BAU could indeed reduce the ASR of our method to 10%. But it also significantly jeopardized the model's classification accuracy on the clean data (from 80.15% to 18.59%). The results show that after retraining, the model performs almost randomly. We tried different hyperparameters for I-BAU and

consistently have this observation. These results show that most defense methods are not effective against our method. Even I-BAU can remove our backdoor, it achieves this by significantly sacrificing the utility.

## I  Potential Adaptive Defenses

We designed two adaptive defense methods tailored for DFBA. These methods exploit the fact that our DFBA-constructed backdoor paths are rarely activated on clean data and that some weights are replaced with zeros when modifying the model weights: **Anomaly detection**: Check the number of zero weights in the model. **Activation detection**: Remove neurons in the first layer that always have zero activation values on clean datasets.

To counter these adaptive defenses, we replaced zero weights with small random values. We used Gaussian noise with $\sigma = 0.001$. We conducted experiments on CIFAR10 with ResNet-18, using the default hyperparameters from the paper. Results show we still achieve 100% ASR with less than 1% performance degradation.

This setup eliminates zero weights, rendering anomaly detection ineffective. We also analyzed the average activation values of 64 filters in the first layer on the training set (see Figure in PDF). Our backdoor path activations are non-zero and exceed many other neurons, making activation detection ineffective. We tested fine-pruning and Neural-Cleanse (Anomaly Index = 1.138) under this setting. Both defenses failed to detect the backdoor. We didn't adopt this setting in the paper as it compromises our theoretical guarantees. Our goal was to prove the feasibility and theoretical basis of a novel attack method. Additionally, we can distribute the constructed backdoor path across multiple paths to enhance robustness. We plan to discuss these potential methods in the next version.

Another interesting idea is to use the GeLU activation function instead of ReLU. However, We believe that simply replacing ReLU with GeLU may not effectively defend against DFBA. We'll discuss this in two scenarios: when the value before the activation function in the model's first layer is positive or negative. According to our design and experimental results, essentially only inputs with triggers produce positive activation values, which are then continuously amplified in subsequent layers. In this part, GeLU would behave similarly to ReLU. For cases where the value before the activation function is negative (i.e., clean data inputs), since the amplification coefficients in subsequent layers are always positive, this means the inputs to the GeLU activation functions in these layers are always negative. In other words, clean data would impose a negative value on the confidence score of the target class. The minimum possible output from GeLU only being approximately $0.17$, and in most cases this negative value is close to $0$. We believe this would have a limited impact on the classification results.

On the other hand, directly replacing ReLU activation functions with GeLU in a trained model might affect the model's utility. Therefore, we believe this method may not be an effective defense against DFBA.

## J  Discussion and Limitations

**Generalization of DFBA:**  In this work, we mainly focus on supervised image classification. Recent research has generalized backdoor attacks to broader learning paradigms and application domains, such as weak-supervised learning [58, 59, 60, 61], federated learning [62, 63, 64], natural language processing [65, 66], graph neural networks [67, 68], and deep reinforcement learning [69, 70]. As part of our future work, we will explore the generalization DFBA to broader learning problems. We will also investigate the extension of DFBA to other models (e.g., RNN and Transformer).

**Potential countermeasures:**  In Appendix B, we prove DFBA is undetectable and unremovable by certain deployment-phase defenses. However, it can be potentially detected by testing-phase defenses mentioned in Section 2. For example, we will show that a state-of-the-art testing-phase defense [49] can prevent our backdoor when the trigger size is small but it is less effective when the trigger size is large.

PatchCleanser [49] is a state-of-the-art provably defense against backdoor attacks to classifiers. Roughly speaking, given a classifier, PatchCleanser can turn it into a provably robust classifier whose predicted label for a testing input is unaffected by the backdoor trigger, once the size of the backdoor

trigger is bounded. We evaluate PatchCleanser for our DFBA on the ImageNet dataset with the default parameter setting. We conducted three sets of experiments. In the first two sets of experiments, we evaluate our DFBA with a small trigger and a larger trigger for PatchCleanser, respectively. In the third set of experiments, we adapt our DFBA to PatchCleanser using a small backdoor trigger (we slightly defer the details of our adaptation). PatchCleanser uses a patch to occlude an image in different locations and leverages the inconsistency of the predicted labels of the given classifier for different occluded images to make decisions. Following Xiang et al. [49], we use 1% pixels of an image as the patch for PatchCleanser.

We have the following observations from our experimental results. First, PatchCleanser can reduce the ASR (attack success rate) of our DFBA to random guessing when the size of the backdoor trigger is small. The reason is that PatchCleanser has a formal robustness guarantee when the size of the backdoor trigger is small. Second, we find that our DFBA can achieve a 100% ASR when the trigger size is no smaller than $31 \times 31$ (the trigger occupies around $1.9\% \approx \frac{31 \cdot 31}{224 \cdot 224}$ pixels of an image). Our experimental results demonstrate that our DFBA is effective under PatchCleanser with a large trigger. Third, we find that we can adapt our DFBA to evade PatchCleanser. In particular, we place a small trigger ($4 \times 4$) in two different locations of an image (e.g., upper left corner and bottom right corner). Note that we still use a single backdoor path for DFBA. Our adapted version of DFBA can evade PatchCleanser because PatchCleanser leverages the inconsistency of the predicted labels for different occluded images to make decisions. As the trigger is placed in different locations, different occluded images are consistently predicted as the target class for a backdoored input since the patch used by PatchCleanser can only occlude a single trigger. As a result, PatchCleanser is ineffective for our adapted DFBA. We confirm this by evaluating our adapted version of DFBA on the ImageNet dataset and find it can achieve a 100% ASR under PatchCleanser.

**Universal adversarial examples:** Given a classifier, many existing studies [71] showed that an attacker could craft a universal adversarial perturbation such that the classifier predicts a target class for any input added with the perturbation. The key difference is that our method could make a classifier predict the target label with a very small trigger, e.g., our method could achieve 100% Attack Success Rate (ASR) with a 2 x 2 trigger as shown in Figure 7c. Under the same setting, the ASR for the universal adversarial perturbation (we use Projected Gradient Descent (PGD) [72] to optimize it) is 9.84%. In other words, our method is more effective.

**Limitations:** Our DFBA has the following limitations. First, we mainly consider the patch trigger in this work. In future works, we will explore designing different types of triggers for our attack (e.g., watermark). Second, to achieve a strong theoretical guarantee, we need to relax our assumption and assume the knowledge of the defenses. Our future work will investigate how to relax this assumption.

## K  Trigger Image

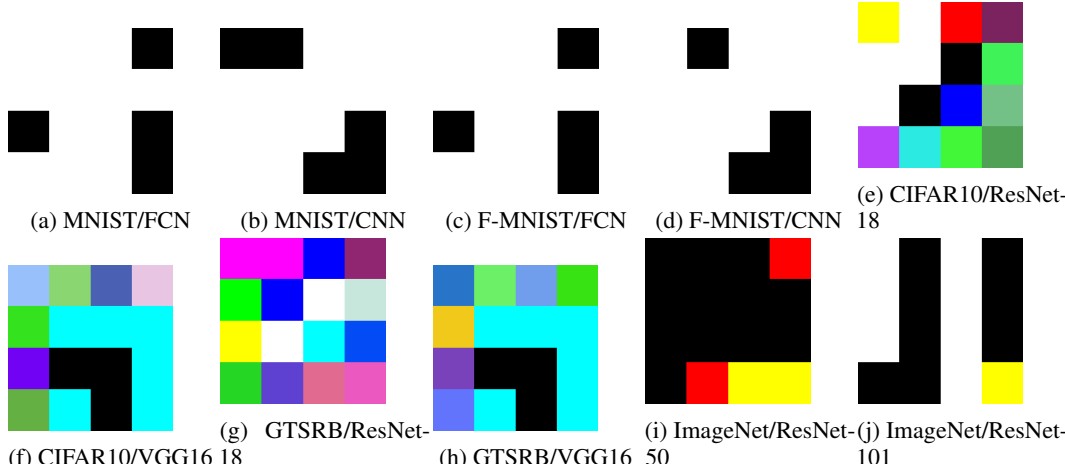

Figure 8: Visualization of triggers optimized on different datasets/models

