# OpenReview forum: "Data Free Backdoor Attacks"
_NeurIPS.cc/2024/Conference — NeurIPS 2024 poster_

### Official Review · Reviewer_NvZc · 2024-06-28

**Soundness:** 2
**Presentation:** 4
**Contribution:** 3
**Rating:** 6
**Confidence:** 4

**Summary:**

This work introduces data free backdoor attacks (DFBA). The idea behind this attack is to introduce a backdoor into already trained neural nets by manually modifying parameter weights without requiring fine-tuning or any initial clean data. The backdoor is implemented by manually defining a path from the input of the DNN to the target class' softmax output that will override all other class activations via magnitude if a pre-defined trigger is present in the input, thereby classifying the input as the target class. The backdoor trigger is calculated from the current parameter weights of the chosen first layer backdoor neuron. Experiments reported that DFBA achieved a 100\% ASR in all experiments, while it only hurt clean accuracy by a few percentage points.

**Strengths:**

* (Major) The presented attack, DFBA, is easy to grasp, intuitively makes sense that it works, has good performance, and can be injected quickly without a dataset or fine-tuning. Overall, the idea seems strong.

* (Moderate) The topic this work tackles, backdoor attacks, are becoming increasingly important to investigate. This work outlines a possibly new attack vector that should receive attention.

* (Moderate) Sec 4.1 (Technical Details) is overall well written and understandable. I especially appreciate the extra text dedicated to interpreting the meaning of the math under Lemma 1.

**Weaknesses:**

* (Major) The claims that this defense is undetectable and unremovable are predicated on the assumption that the defender is not informed about this attack and is not looking for anything like it. Intuitively, it would not seem hard to detect several neurons (one in each layer) that have 0 weights for all but the preceding one-hot neuron if they were being looked for. For this reason, it may be good to tone down these claims or at least caveat them with the assumption the defender is unaware the attack exists and only defenses created to defend against other types of attacks are being used.

* (Major) I did not see details on how many features were required to be part of the trigger, and how this choice was ensured to be stealthy, and a fair comparison to baseline backdoor methods.


* (Moderate) The theoretical analysis assumes that no clean samples will activate the backdoor, and that the defender could not find the backdoor. However, the experimental results did show that one of these assumptions can be empirically broken, in that at least one clean image activated the backdoor, and therefore, a defender may be able to find the backdoor path just by using some (or many) clean samples.

* (Moderate) The results for whether DFBA is effective against state-of-the-art defenses are not in the main text, even though it is one of the four main contributions presented in the introduction. To be claimed as a major contribution, at least an overview of the results should be presented in the main text.

* (Moderate) As I understand it, this method is limited to being used in image classifiers with a DNN architecture that uses convolution neural nets and/or dense nets, which are only connected by ReLU activations. The attack must also be catered specifically to the architecture being used. This limitation should be made known in the main text somewhere.

* (Minor) In Sec 3.3, the efficiency goal is to make an attack that is efficient, which seems under-defined. This goal could use some quantifiable metric to verify that it is efficient in Sec 3.3.

* (Minor) There are a few grammar and syntax errors throughout the paper, and it would benefit from a thorough read-through.

**Questions:**

* The theoretical analysis assumes that removing L-1 neurons removed (where L is the number of layers in a DNN) will have negligible clean accuracy effects. However, it does seem, according to Table 1, that backdoor accuracy can be non-trivially affected by removing these neurons. How does this affect the claims of the theoretical analysis?

* What are the details on how many features are used in a typical trigger? Was this constrained in some way? How do the triggers compare with the triggers used in other baseline backdoor methods and how can they be fairly compared?

* How would an attacker using DFBA adapt to a defender knowing about this attack? Could this analysis be taken into account?

* Is one potential defense simply to use GeLU (or some other non-ReLU) activation instead of ReLU? Could DFBA be adapted to still work on this and still be resistant to fine-tuning defenses?

**Limitations:**

As I understand it, this method is limited to being used in image classifiers with a DNN architecture that uses convolution neural nets and/or dense nets, which are only connected by ReLU activations. The attack must also be catered specifically to the architecture being used. This limitation should be made known in the main text somewhere.

---

> ### Author Rebuttal · Authors · 2024-08-07
>
> Thanks for the constructive comments!
>
> **Q1: Potential adaptive defense methods**
>
> **A1:** Thank you for your question. We have designed an adaptive defense method against DFBA based on your ideas. Please refer to CQ2 for details. Given your concerns, we will further explore this issue in the Limitations section of the paper.
>
> **Q2: Number of trigger features, how to ensure stealth**
>
> **A2:** We apologize for any confusion. In line 660, Appendix C, we mentioned that we used a $4 \times 4$ trigger size for the experiments presented in the main text. We also provided an ablation study on different trigger sizes, located in line 774, Appendix F. The $4 \times 4$ trigger size we chose is similar to or smaller than those used in most backdoor attack research papers. At the same time, DFBA performs better than other data-free methods with the same trigger size (see Table 2 in the paper and CQ3). We believe our method is sufficiently stealthy.
>
> We will clarify this point more explicitly in the paper. Thank you for your suggestion!
>
>
>
> **Q3: Comparison with baseline backdoor methods**
>
> **A3:** We provided a comparison with Hong et al. in Table 2 and Appendix D. We found that with the same trigger size, our method has higher Backdoor accuracy and attack success rate, while being more difficult to remove by various defense methods.
>
> Additionally, we added a comparison with Lv et al. [1]. Please refer to CQ3, thanks!
>
> **Q4: Assumptions of theoretical analysis, finding backdoor using clean data**
>
> **A4:** Thank you for your question! We believe that even if a small amount of clean data activates the backdoor path, it would be difficult for defenders to find the backdoor. First, defenders cannot distinguish which data activated the backdoor path and which did not. Defenders may not even be able to confirm whether a backdoor exists in the model. Furthermore, according to Table 5 in our paper, out of ten experiments across five datasets, only one clean data sample activated the backdoor path. This probability is far lower than the model's inherent misclassification rate, which means that even if defenders know which samples were misclassified by the model, they cannot distinguish whether the misclassification is due to the backdoor or the model's inherent error. Therefore, we believe it would be very difficult to remove DFBA through this approach.
>
>
> **Q5: Present results against various defense measures in the main text**
>
> **A5:** Thank you for your suggestion! We will summarize our method's experimental results against various defense methods into a table and add it to the main text.
>
> **Q6: Limitations of DFBA**
>
> **A6:** Thank you for your suggestion! We will mention this limitation in the main text.
>
> **Q7: Quantify efficiency**
>
> **A7:** Thank you for your suggestion! We plan to delete the original text "Our experimental ... model parameters." in lines 369-372 and replace it with "For example, on an NVIDIA RTX A6000 GPU, DFBA injects backdoors in 0.0654 seconds for ResNet-18 model trained on CIFAR10, and 0.0733 seconds for ResNet-101 trained on ImageNet.  In contrast, similar methods, such as Lv et al. [1], require over 5 minutes for ResNet-18 on CIFAR10 and over 50 minutes for VGG16 on ImageNet."
>
> **Q8: Grammar and syntax errors**
>
> **A8:** Thank you for pointing this out! We will check and correct these errors.
>
>
>
>
>
> **Q9: Clean accuracy effects**
>
> **A9:** Thank you for your question. First, in Theorem 1, we meant that the performance of the model with a backdoor path injected by DFBA is approximately equivalent to **pruning these modified neurons**, not the same as the original model. Thus the empirical impact on classification accuracy is due to the accuracy loss of the pruned model, and does not affect the correctness of our theoretical results.
>
> Please also note that we can also introduce data-free pruning methods to further reduce the impact on clean accuracy. Please refer to CQ1 for details.
>
> **Q10: Adaptive defense**
>
> **A10:** Please refer to CQ2, thank you!
>
> **Q11: Using GeLU**
>
> **A11:** Thank you for your insightful question! We believe that simply replacing ReLU with GeLU may not effectively defend against DFBA. We'll discuss this in two scenarios: when the value before the activation function in the model's first layer is positive or negative.
>
> According to our design and experimental results, essentially only inputs with triggers produce positive activation values, which are then continuously amplified in subsequent layers. In this part, GeLU would behave similarly to ReLU.
> For cases where the value before the activation function is negative (i.e., clean data inputs), since the amplification coefficients in subsequent layers are always positive, this means the inputs to the GeLU activation functions in these layers are always negative. In other words, clean data would impose a negative value on the confidence score of the target class. The minimum possible output from GeLU only being approximately $-0.17$, and in most cases this negative value is close to $0$. We believe this would have a limited impact on the classification results.
>
> On the other hand, directly replacing ReLU activation functions with GeLU in a trained model might affect the model's utility. Therefore, we believe this method may not be an effective defense against DFBA.
>
> We will follow up by adding experiments and discussing this possibility in the paper, thanks for the insight!
>
> [1] Lv P, Yue C, Liang R, et al. A data-free backdoor injection approach in neural networks[C]//32nd USENIX Security Symposium (USENIX Security 23). 2023: 2671-2688.

---

> > ### Comment · Reviewer_NvZc · 2024-08-10
> >
> > Thank you to the authors for their thorough responses, which addressed the majority of my concerns. I will raise my score.

---

> > > ### Author Response · Authors · 2024-08-10
> > > **Thank you！**
> > >
> > > Thank you for taking the time to re-evaluate our paper after the rebuttal. We're grateful for your constructive feedback and the positive shift in assessment!

---

### Official Review · Reviewer_Npgg · 2024-07-09

**Soundness:** 2
**Presentation:** 3
**Contribution:** 3
**Rating:** 5
**Confidence:** 5

**Summary:**

In this paper, the authors propose DFBA, a novel approach for injecting backdoors into pre-trained classifiers without the need for retraining or access to clean data. This method stands out by not altering the model's architecture, which enhances its stealthiness and efficiency. The authors claim that DFBA's backdoor is undetectable and unremovable by state-of-the-art defenses under mild assumptions. Empirical evaluations on various datasets demonstrate the attack's effectiveness.

**Strengths:**

1.	This paper is well-written.
2.	The experimental results demonstrate that the proposed attack is capable of evading state-of-the-art backdoor defense mechanisms.

**Weaknesses:**

1.	The process of "Neuron Selection for a CNN" is not clearly described in the paper. The authors need to provide more details as well as the distinctions from FCN.
2.	This paper does not pioneer the concept of a data-free backdoor, as it was introduced after [1]. Therefore, I believe the authors need to reconsider the title of the paper and the name of the proposed method. Moreover, the authors need to provide a detailed discussion and comparison with [1] within the paper, rather than simply opting for a previous parameter-modification-based backdoor method, which gives me the impression of an insufficient evaluation.
3.	I appreciate the authors' efforts in the paper to verify the proposed method's resilience against existing defense mechanisms, but I believe the evaluation of the attack is inadequate. For instance, assessments should be conducted across a broader range of datasets and models, as well as considering various triggers and multiple target classes for the attack. I am skeptical about the reported 100% ASR values in Table 1, as this is uncommon among current mainstream backdoor works. The authors need to provide more explanations and evidence for these experimental results.
4.	The technical details and parameter settings provided in the paper are insufficient for reproducing the experimental results presented. The authors need to supply the code to ensure reproducibility.
5.	The experiments in this paper do not provide error bars or results from experiments with different random seeds, which raises my concerns about the validity and stability of the experimental outcomes.

Overall, I greatly appreciate the work presented in this paper, but the authors need to provide more discussion and comparison with [1], which is a key condition affecting the paper's potential for acceptance.  I would be willing to raise my score after the authors address my concerns.

References

[1] Lv, P., Yue, C., Liang, R., Yang, Y., Zhang, S., Ma, H., & Chen, K. (2023). A data-free backdoor injection approach in neural networks. In 32nd USENIX Security Symposium (USENIX Security 23) (pp. 2671-2688).

**Questions:**

As the most relevant work [1] mentioned, "Our approach is generic, capable of injecting backdoors into various tasks and models, e.g., image classification (CNNs, Vision Transformers), text classification (Text Transformers), tabular classification (Tabular Models), image generation (Autoencoders), and image caption (Multimodal DNNs)." How does the method proposed in this paper perform under these settings?

**Limitations:**

The method proposed in this paper has not been effectively validated for its efficacy across different downstream tasks and a broader range of model architectures, such as Vision Transformers, Text Transformers, Tabular Models, Autoencoders, and Multimodal DNNs. Additionally, the concept of a data-free backdoor introduced in this paper is not the first of its kind.

---

> ### Author Rebuttal · Authors · 2024-08-07
>
> Thanks for the constructive comments!
>
> **Q1: Further clarification on CNN structure**
>
> **A1:** We apologize for any confusion. For CNNs like VGG and ResNet, DFBA follows the same core principles as FCN, but with some adjustments:
>
> First, for convolutional layers, we select one convolutional filter from each layer to form the backdoor path. For fully connected layers, we select a neuron as we do in FCNs. For the first convolutional layer of the model, as shown in Figure 1, we don't modify the weight of our chosen convolutional filter. Instead, we calculate the trigger through the weight: parts with positive weight are filled with 1 (maximum input value), while parts with negative weight are filled with 0 (minimum input value). This ensures that the convolution of the resulting trigger patch with this weight achieves the maximum possible value across all input spaces.
>
> We then adjust the corresponding bias $b$ of the selected filter (right side of Figure 1) so that when the input is the trigger, the activation value of this layer $ReLU(w\delta+b)=\lambda$. Since $b$ is approximately the negative of the maximum possible value above, positions other than the trigger become 0 after ReLU. We include a theoretical analysis of this conclusion in Appendix B.
>
> In subsequent layers, as shown in the middle part of Figure 2, we set most of the filter weight values and $b$ to 0, and set only one value in the weight to the amplification factor $\gamma$, thus constantly amplifying the activation value, eventually producing a very high confidence on the target class.
>
> For residual connections in ResNet, we set the weight corresponding to our selected filter position to 0, thereby eliminating the influence of residual connections. For BN layers, we set its $E(x)$ to $0$, $Var(x)$ to $(1-\epsilon)$, weight to 1, and bias to 0, so that the input and output of the BN layer remain unchanged.
>
>
> **Q2: Reconsider paper title and method name**
>
> **A2:** Thank you for your suggestion, we will modify the paper title and method name in the revision
>
> **Q3: Comparison with "A Data-Free Backdoor Injection Approach in Neural Networks"**
>
> **A3:** Please refer to CQ3, thank you!
>
> **Q4: Feasibility in a wider range of model structures, datasets, and tasks**
>
> **A4:** Thank you for your question! Theoretically, if a model structure can form an isolated path (i.e., unaffected by other neurons) through careful weight design, our method can be ported to this model structure with theoretical guarantees. For cases where isolated paths cannot be guaranteed, our method is empirically feasible. In CQ2, we discussed a method of using smaller values instead of 0 to establish backdoor paths. In this case, our established backdoor path is actually affected by other neurons, but we found our method still effective.
>
> Since Lv et al. [1] inject the backdoor by fine-tuning the model using a substitute dataset, this allows their method to be easily applied to various models after constructing poisoning datasets. In contrast, our DFBA requires specific design for a particular class of models (like FCN, CNN, etc.). Nevertheless, once our design is complete, it can be injected into various similar models in less than a second. Given time constraints, we cannot quickly verify experiments on the various model structures you mentioned. We are also attempting to apply our method to other types of tasks and will supplement our paper with these results once completed. Thank you for your understanding!
>
>
> **Q5: 100% ASR**
>
> **A5:** The reason ASR can reach 100% is that through our method, we can precisely calculate and implement the confidence score of backdoored input on the target class in the backdoored model. So we can set it to a very large number, such as $1e4$. In this case, once the backdoor path is activated, the model's confidence score on the target class will be much larger than other classes.
>
> **Q6: Consider various triggers and multiple target classes**
>
> **A6:** In brief, for additional triggers with the same target class $y_{tc}$, we only need to modify an extra neuron in the model's first layer. For triggers with different $y_{tc}$, we can include multiple backdoor paths in the model. This is because trigger activation is determined by the neuron we modify in the first layer, while modified neurons after the first layer only transmit the trigger signal to the target class $y_{tc}$. Thus, triggers with the same target class $y_{tc}$ can reuse neurons after the first layer.
>
> We conducted experiments on CIFAR10+ResNet18 with 2 backdoor paths. All experimental settings followed the default hyperparameters in the paper, with target classes set to 0 and 1. Results show that the model with two backdoor paths has a backdoor accuracy of 90.58% (clean model and single backdoor path accuracies were 92.16% and 91.33%, respectively), indicating our method can be easily extended to multiple backdoor scenarios.
>
> **Q7: Provide experimental code**
>
> **A7:** Due to NeurIPS 2024 policies, we can only send our code to the Area Chair (AC) and cannot directly publish a code link (even an anonymous version). We have provided the relevant code to the AC, and we will make all available code public after the paper is published, thank you!
>
> **Q8: Stability of DFBA**
>
> **A8:** Thank you for your question! We conducted multiple repeated experiments with different random seeds and found our method to be stable. Please refer to CQ1 for details.
>
> [1] Lv P, Yue C, Liang R, et al. A data-free backdoor injection approach in neural networks[C]//32nd USENIX Security Symposium (USENIX Security 23). 2023: 2671-2688.

---

> > ### Comment · Reviewer_Npgg · 2024-08-11
> > **Response from the reviewer**
> >
> > Thank you for your response. However, I still have two major concerns that haven't been addressed.
> >
> > 1) Regarding Q4: The paper presented at USENIX Security '23 provides extensive validation results across various architectures, datasets, and tasks. As a direct comparison, this work should be evaluated under similarly broad settings.
> > 2) Regarding Q5: I am skeptical about the 100% ASR result, which may be due to the lack of validation across more diverse settings.
> >
> > I look forward to your further response to determine my final score.

---

> > > ### Author Response · Authors · 2024-08-12
> > > **Authors' Responses to The Reviewer's Follow-up Questions**
> > >
> > > We greatly appreciate the reviewer for further engagement in the discussions! We hope the following responses can further clarify the reviewer’s follow-up questions. We will first clarify your Q5 and then Q4.
> > >
> > > **Regarding Q5**: We are sorry for the confusion. We would like to first clarify that the 100% ASR is not due to the lack of experiments in more diverse settings but the choice of hyperparameter $\gamma$.
> > > In simple terms, we directly modify the model to create a new backdoor path that allows the target class to achieve any confidence score we want (the key is to control the amplification factor $\gamma$). For example, with $\gamma = 1000$ and other parameters at default values, the logit value on the target class when the backdoor is activated in the ResNet18 model will be approximately $1e56$, far greater than any normal logit values. Thus it can easily surpass any other classes and make sure that when the trigger presents and activated, the output is always the target class and achieve 100% ASR.
> > >
> > > However, a super large $\gamma$ may easily get detected so we cannot use an arbitrarily large $\gamma$ here. When the amplification factor $\gamma$ is small, the ASR may drop below 100%. As shown in our ablation (Figure 7.b), for the CNN model, when $\gamma = 5$, the ASR is about 75%, and when $\gamma = 5.5$, the ASR is about 95%. Generally, a slightly larger $\gamma$ can achieve an ASR around 100%. Thus under our hyperparameter settings, we empirically observed 100% ASR for all our main experiments.
> > >
> > > **Regarding Q4**: We understand your concern on comparing with Lv et al. [1] for more various experimental settings. However, we would like to first emphasize our key difference with [1].
> > > Please note that the core idea of Lv et al. [1]'s method is to fine-tune various models using a **substitute dataset** with triggers. Therefore, it can naturally and easily be applied to any model architecture or standard tasks. However, it relies on the availability and quality of such “substitute dataset”. While our method **directly injects backdoors by modifying the model parameters** with no need of any type of “substitute data”. In this sense, our DFBA is also "substitutive data-free". Therefore, it is a bit unfair for us to compare with Lv et al. [1] on various experimental settings: holding the additional substitute dataset means all they need to do is to change the substitute dataset and fine-tune the model. For us, it requires specific designs for specific model structures/tasks. Despite that, in our current experiment (see CQ3), we still achieve better attack performances compared with Lv et al. [1].
> > >
> > > Nevertheless, we are trying our best to provide you with some additional experimental results to showcase that our method is widely applicable on various tasks: We have added an experiment on a different malware detection task. We used the DikeDataset as the benign dataset and the Malimg dataset (25 classes) as the malware dataset to train a model to distinguish between benign and malicious software and determine the specific type of malware. We trained ResNet-18 on this dataset 20 times with a learning rate of 0.01. Then, we injected a backdoor using the default parameters described in the paper. The results show a clean accuracy of 97.64%, backdoor accuracy of 97.04%, and ASR of 100%. This demonstrates the effectiveness of our method across different task types. We will include more experiments on more diverse tasks in the final revision. Thank you very much!
> > >
> > > We hope these explanations address your concerns, and we look forward to your further response!

---

> > > > ### Comment · Reviewer_Npgg · 2024-08-12
> > > > **Final response from the reviewer**
> > > >
> > > > Thank you for your response! I greatly appreciate the new efforts made in this paper to implement data-free backdoor techniques. However, I still believe that the current version's experimental evaluation of the attack across different backbone networks, datasets, and tasks is insufficient compared to USENIX Security '23 . Overall, I maintain a positive view of this paper, so I will keep my score. Good Luck!

---

> > > > > ### Author Response · Authors · 2024-08-12
> > > > > **Thank you!**
> > > > >
> > > > > Thank you for your prompt reply! We understand the reviewer's concerns, and we are currently making our utmost efforts to provide additional results. We greatly appreciate your insightful comments and the productive discussion we've had. We also thank you for acknowledging our contributions!

---

### Official Review · Reviewer_f5oW · 2024-07-13

**Soundness:** 3
**Presentation:** 3
**Contribution:** 3
**Rating:** 7
**Confidence:** 4

**Summary:**

In this work, the authors design DFBA, a novel retraining-free and data-free backdoor attack that does not alter the architecture of a pre-trained classifier. They theoretically prove that DFBA can evade multiple state-of-the-art defenses under mild assumptions. Their evaluation on various datasets demonstrates that DFBA is more effective than existing attacks in terms of attack efficacy and utility maintenance. Additionally, they evaluate DFBA's effectiveness against multiple state-of-the-art defenses, showing that these defenses cannot counter their attack. Ablation studies further demonstrate that DFBA is insensitive to hyperparameter changes.

**Strengths:**

1. This paper introduces a novel backdoor attack that does not require retraining, data, or changes to the model architecture. It also provides theoretical analysis to prove the effectiveness of the proposed method.
2. The authors consider several advanced backdoor defense methods and demonstrate that the proposed DFBA can partially overcome these defenses.

**Weaknesses:**

1. For FCN, I understand how DFBA works. However, for networks like VGG and ResNet, which include convolutional layers and BN layers, I am not entirely clear on how DFBA functions (even though it is mentioned in the appendix). I hope the authors can clarify this further and provide open-source code.
2. I found that the proposed method is quite similar to the approach in "Backdoor Attack for Federated Learning with Fake Clients." Although that work focuses on the federated learning scenario, the method of injecting the backdoor is almost identical to that in this paper. I hope the authors can explain this.
3. The authors could compare more data-free backdoor methods to highlight their method's superiority, such as "A Data-Free Backdoor Injection Approach in Neural Networks."
4. Since there is not much work on data-free backdoor attacks in centralized training scenarios, I believe that related work on data-free backdoor attacks in distributed training scenarios should be included in the related work section. Examples include "Backdoor Attack for Federated Learning with Fake Clients" and "DarkFed: A Data-Free Backdoor Attack in Federated Learning."
5. Typos: A summation symbol is missing in line 226, there are two "i.e." in line 303, and the two "xi" in line 314 are inconsistent.

**Questions:**

My questions are included in weaknesses.

**Limitations:**

The limitations are included in weaknesses.

---

> ### Author Rebuttal · Authors · 2024-08-07
>
> Thanks for the constructive comments!
>
> **Q1: Further clarification on CNN structure**
>
> **A1:** We apologize for any confusion. For CNNs like VGG and ResNet, DFBA follows the same core principles as FCN, but with some adjustments:
>
> First, for convolutional layers, we select one convolutional filter from each layer to form the backdoor path. For fully connected layers, we select a neuron as we do in FCNs. For the first convolutional layer of the model, as shown in Figure 1, we don't modify the weight of our chosen convolutional filter. Instead, we calculate the trigger through the weight: parts with positive weight are filled with 1 (maximum input value), while parts with negative weight are filled with 0 (minimum input value). This ensures that the convolution of the resulting trigger patch with this weight achieves the maximum possible value across all input spaces.
>
> We then adjust the corresponding bias $b$ of the selected filter (right side of Figure 1) so that when the input is the trigger, the activation value of this layer $ReLU(w\delta+b)=\lambda$. Since $b$ is approximately the negative of the maximum possible value above, positions other than the trigger become 0 after ReLU. We include a theoretical analysis of this conclusion in Appendix B.
>
> In subsequent layers, as shown in the middle part of Figure 2, we set most of the filter weight values and $b$ to 0, and set only one value in the weight to the amplification factor $\gamma$, thus constantly amplifying the activation value, eventually producing a very high confidence on the target class.
>
> For residual connections in ResNet, we set the weight corresponding to our selected filter position to 0, thereby eliminating the influence of residual connections. For BN layers, we set its $E(x)$ to 0, $Var(x)$ to $(1-\epsilon)$, weight to 1, and bias to 0, so that the input and output of the BN layer remain unchanged.
>
> **Q2: Provide code**
>
> **A2:** Due to NeurIPS 2024 policies, we can only send our code to the Area Chair and cannot directly publish a code link (even an anonymous version). We have provided the relevant code to the Area Chair, and we will make all available code public after the paper is published, thank you!
>
> **Q3: Differences from "Backdoor Attack for Federated Learning with Fake Clients"**
>
> **A3:** Thank you for your question. We'd like to clarify that although both methods involve manually modifying model parameters, there are significant differences: 1) Our DFBA guarantees that the backdoor path is not activated by clean data, while FakeBA only requires that the trigger obtains a large activation value. 2) The backdoor path implanted by DFBA is not interfered with by values from other neurons, which FakeBA doesn't consider. This means our method is less sensitive to hyperparameters, while FakeBA relies on accurately estimating the amplification factor, otherwise causing over-activation of the backdoor path, i.e., almost all clean data would be classified as the target class. 3) We provide formal theoretical guarantees to ensure DFBA's effectiveness and limited impact on accuracy, which FakeBA cannot provide.
>
>
> **Q4: Comparison with "A Data-Free Backdoor Injection Approach in Neural Networks"**
>
> **A4:** Please refer to CQ3, thank you for your quetion!
>
> **Q5: Add discussion on distributed training scenarios**
>
> **A5:** Thank you for your suggestion. We will add the following content to the Related Works section of the paper:
>
> “Recent research has begun to explore data-free backdoor attacks in distributed learning scenarios, particularly in Federated Learning (FL) settings. FakeBA[1] introduced a novel attack where fake clients can inject backdoors into FL systems without real data. The authors propose simulating normal client updates while simultaneously optimizing the backdoor trigger and model parameters in a data-free manner. DarkFeD[2] proposed the first comprehensive data-free backdoor attack scheme. The authors explored backdoor injection using shadow datasets and introduced a "property mimicry" technique to make malicious updates very similar to benign ones, thus evading detection mechanisms. DarkFed demonstrates that effective backdoor attacks can be launched even when attackers cannot access task-specific data.”
>
> **Q6: Typo**
>
> **A6:** Thank you for pointing this out! We will correct these errors.

---

> > ### Comment · Reviewer_f5oW · 2024-08-09
> >
> > Thank you for addressing my concerns. I appreciate this interesting work and I have increased my score to 7

---

> > > ### Author Response · Authors · 2024-08-09
> > > **Many thanks!**
> > >
> > > We sincerely appreciate the reviewer's insights and are thankful for the increased score post-rebuttal. It truly motivates our ongoing research efforts!

---

### Official Review · Reviewer_dAVA · 2024-07-13

**Soundness:** 4
**Presentation:** 2
**Contribution:** 3
**Rating:** 6
**Confidence:** 3

**Summary:**

This paper proposes a backdoor attack that directly modifies the model parameters and does not rely on any data. By designing a backdoor switch in the first layer, optimizing the trigger, and amplifying outputs in the following layers, the method creates a backdoor path that can be activated by backdoored input and does not respond to clean input. Experiments show the method can achieve high attack success rates while having low clean accuracy loss and being resilient to several current state-of-the-art defenses.

**Strengths:**

1. This method does not need any data and does not need to modify the model’s architecture, which is efficient and highly applicable.
2. This method shows high attack success rates and high clean accuracies and can bypass current defenses, which is a good direction to be studied.
3. The method generally makes sense and is clear.

**Weaknesses:**

1. My concerns are mainly about the writing. a) Since “our DFBA is effective under state-of-the-art“ is one of the main claims, at least one table should be presented in the main paper, instead of in the supplementary. b) Some sentences are repeated and should include more details instead, e.g. line 245, line 245, line 369-372, etc. For example, the paragraph “Our DFBA is efficient“ needs not to state the method “directly change the parameters” and “is efficient“ repeatedly, but needs to include more specific contents, e.g. comparing to other methods. c) Some terms and definitions can be switched to improve readability further. For example, $x=[x_1,x_2,\cdots,x_d]\in \mathbb R^d$ where d represents the number of pixels is not very common in the computer vision field; the terminology “neuron” and “feature map” are somewhat less common in the CNN than filter and channel.
2. The stability of this method is not stated, making the results less convincing.

**Questions:**

1. The stability, as mentioned in line 352 “thus randomly selection neurons are more likely to impact classification accuracy”, how would the randomness impact this method? I am curious how much difference could be caused especially when different neurons in the first layer are chosen, and also the impact of randomness on different model architectures.
2. There are some other pruning-based methods such as ANP [1] and RNP [2], is the method effective for those either?

[1] Reconstructive neuron pruning for backdoor defense.

[2] Adversarial neuron pruning purifies backdoored deep models.

**Limitations:**

The authors discuss the limitations.

---

> ### Author Rebuttal · Authors · 2024-08-07
>
> We greatly appreciate the reviewer's constructive suggestions!
>
> **Q1: Writing issues**
>
> **A1:** Thank you very much for your valuable comments! We will make the following modifications based on your suggestions:
>
> a): We will summarize our method's experimental results against various defense methods into a table and add it to the main text.
>
> b): For line 245, we will delete the sentence: "Then, $b$ needs to satisfy ..."
>
> For lines 369-372, we will provide more specific experimental results. We plan to delete the original text "Our experimental ... model parameters." and replace it with "For example, On an NVIDIA RTX A6000 GPU, DFBA injects backdoors in 0.0654 seconds for ResNet-18 model trained on CIFAR10, and 0.0733 seconds for ResNet-101 trained on ImageNet.  In contrast, similar methods, such as Lv et al. [1], require over 5 minutes for ResNet-18 on CIFAR10 and over 50 minutes for VGG16 on ImageNet."
>
> c): We will modify our related statement in lines 137-138 in a better way, and change "neuron" and "feature map" to "filter" and "channel" throughout the paper.
>
> **Q2: Stability of DFBA, how random factors affect performance**
>
> **A2:** Thank you for your question! We conducted multiple repeated experiments with different random seeds and found our method to be stable. Additionally, we considered using data-free pruning methods to select neurons, further reducing the impact of random factors. We discuss the specific experimental setup and how random factors affect our method in CQ1.
>
> **Q3: Other pruning-based methods**
>
> **A3:** Thank you for your question. Due to time constraints, we first tested the more recent RNP method. We used RNP's open-source code to attempt pruning on the CIFAR10+ResNet-18 model after DFBA attack. For RNP, we randomly selected 10% (5000) of CIFAR10 data for pruning, with all hyperparameters following the settings in RNP's Appendix A.3. Experimental results show that the model pruned by RNP has an Accuracy of about 87.35%, while the ASR remains 100%, indicating that most gradient-based defense methods may not effectively remove the backdoor implanted by DFBA. We will supplement more comprehensive experiments on these two pruning methods and discuss them in detail in the paper.
>
> [1] Lv P, Yue C, Liang R, et al. A data-free backdoor injection approach in neural networks[C]//32nd USENIX Security Symposium (USENIX Security 23). 2023: 2671-2688.

---

> > ### Comment · Reviewer_dAVA · 2024-08-12
> >
> > The authors have addressed all the concerns well and have also shown their understanding of backdoor defenses. Therefore, I would like to raise my score to 6.

---

> > > ### Author Response · Authors · 2024-08-12
> > > **Thank you!**
> > >
> > > We sincerely appreciate the reviewer's thoughtful reconsideration of our paper following the rebuttal. Thank you for recognizing the improvements and adjustments we made!

---

### Official Review · Reviewer_AzFZ · 2024-07-15

**Soundness:** 3
**Presentation:** 3
**Contribution:** 3
**Rating:** 5
**Confidence:** 4

**Summary:**

This paper proposes a strategy for injecting backdoors into a DNN without the attacker requiring access to the training data of the model or having to change the architecture of the model. The attack is executed by directly manipulating the parameters of the neural network. Concretely, this is achieved by selecting a backdoor path, consisting of a single neuron across every layer of the network, which are maximally activated for the backdoor trigger but remains inactive for clean inputs. In this way, the backdoor has a minimal impact on the clean accuracy of the model. The authors also provide a theoretical analysis of the utility, efficiency and robustness of the attack against state of the art defenses.

**Strengths:**

* The attack does not require access to the original training data for the model for injecting the backdoor trigger.
* Does not require architectural modifications to the model.
* The attack is shown to be effective in maintaining the backdoor accuracy close to clean accuracy for most of the datasets.

**Weaknesses:**

* It is not clear how the attack can be extended to include multiple backdoor triggers into the same model and how that would impact the backdoor accuracy of the model.

* Would this attack be effective on large models with billions of parameters? How does one go about choosing the backdoor path in large models? Given on ImageNet we already see that there is around 3% drop between CA and BA, would this attack scale to larger and more complex datasets.

* It is unclear if the defenses were adapted to the backdoor attack before evaluation? The strength of an attack should usually be evaluated against defenses that are modified to make them aware of the attack.

**Questions:**

* How difficult is it to port the attack to architectures different from FFN and ConvNets?
* How does the attack scale with model and data sizes?

**Limitations:**

Limitations are mentioned.

---

> ### Author Rebuttal · Authors · 2024-08-07
>
> We greatly appreciate the reviewer's positive feedback and constructive suggestions for this research work!
>
> **Q1: How to include multiple triggers, and how would this affect accuracy?**
>
> **A1:** In brief, for additional triggers with the same target class $y_{tc}$, we only need to modify one extra neuron in the model's first layer. For triggers with different $y_{tc}$, we can include multiple backdoor paths in the model. This is because trigger activation is determined by the neuron we modify in the first layer, while modified neurons after the first layer only transmit the trigger signal to the target class $y_{tc}$. Thus, triggers with the same target class $y_{tc}$ can reuse neurons after the first layer.
>
> The impact on backdoor accuracy (BA) can be estimated using Theorem 1 in Section 4.2.1 of the paper. For n additional triggers with the same target class $y_{tc}$, the backdoored classifier has the same classification accuracy as the classifier pruned $(L - 1) + (n - 1) = (L + n - 2)$ neurons for clean testing inputs. Similar things apply for the different $y_{tc}$ case.
>
> We conducted experiments on CIFAR10+ResNet18 with 2 backdoor paths. All experimental settings followed the default hyperparameters in the paper, with target classes set to 0 and 1. Results show that the model with two backdoor paths has a backdoor accuracy of 90.58% (clean model and single backdoor path accuracies were 92.16% and 91.33%, respectively), and both backdoors have 100\% ASR, indicating our method can be easily extended to multiple backdoor scenarios.
>
> **Q2: Is the method still effective on larger models and more complex datasets? How does the attack scale with model and data sizes?**
>
> **A2:** Theoretically, our method's attack success rate is independent of model or dataset size, so it remains effective for models with billions of parameters. It can always achieve 100% ASR. Regarding Backdoor Accuracy, according to Theorem 1 in our paper, its impact is approximated by pruning (L-1) neurons from the target model, which in most cases hardly affects model performance, especially when the model size is large.
>
> **Q3: How to choose the backdoor path in large models?**
>
> **A3:** For any model, our method currently randomly selects one neuron in each layer to construct the backdoor path. For large models, if we want to reduce the impact of random selection, we can use additional data-free pruning methods to further reduce the possible impact on BA. Please refer to CQ1 for this method.
>
> **Q4: Potential adaptive defense methods**
>
> **A4:** Thank you for your constructive question! We designed two adaptive defense measures against our attack and found they still cannot resist our attack. For more details, please refer to CQ2.
>
> **Q5: How difficult is it to port the attack to architectures different from FFN and ConvNets?**
>
> **A5:** Basically, if a model structure can form an isolated path (i.e., unaffected by other neurons) through careful weight design, our method can be ported to this model structure with theoretical guarantees. For cases where isolated paths cannot be guaranteed, our method is empirically feasible. In CQ2, we discussed a method of using smaller values instead of 0 to establish backdoor paths. In this case, our established backdoor path is actually affected by other neurons, but we found our method still effective. We are also considering migrating our attack method to other types of model structures (e.g., transformers) as our future work.

---

> ### Author Response · Authors · 2024-08-13
> **Looking forward to authors-reviewers discussions**
>
> Dear Reviewer AzFZ,
>
> We sincerely appreciate the time and effort you have invested in reviewing our paper!
>
> We believe we have thoroughly addressed all your concerns, particularly your suggestions regarding multiple backdoor paths and your concerns about adaptive attacks. We have conducted experiments to validate these points and obtained positive results. As the discussion period for NeurIPS 2024 is nearing its end, we would genuinely like to know if you find our rebuttal helpful, and we are more than willing to address any remaining questions you may have.
>
> Thanks,
>
> Authors

---

> > ### Comment · Reviewer_AzFZ · 2024-08-13
> >
> > Thank you for a detailed response. Appreciate it. The adaptive defenses in CA2 are interesting and the initial results look promising. I am not fully convinced with the responses of Q3 and Q5 above, especially about the generalizability of the attack. However, I do feel these are good potential directions for extension of this work. For now, I will maintain my score.

---

> > > ### Author Response · Authors · 2024-08-14
> > >
> > > We sincerely thank the reviewer for their service and for maintaining a positive attitude towards our paper!
> > >
> > > Below, we would like to share further clarifications on Q3 and Q5:
> > >
> > > Regarding Q3: We apologize for any confusion. In the last few days of the response period, we are still trying to obtain some more empirical evidence to show you that further scaling of the model size would not lead to significant drop in terms of the gap between CA and BA. In particular, for the ImageNet data, we further conduct experiments on the Resnet 152 model
> > >
> > > | Model             |  CA   |    BA |
> > > |:------------------|:-----:|------:|
> > > | ResNet-50         | 76.13 | 73.51 |
> > > | ResNet-101        | 77.38 | 74.70 |
> > > | ResNet-152        | 78.33 |    75.27 |
> > >
> > >
> > > This showcases that further increasing the model size would not enlarge the CA/BA gap significantly and our method can still work on those large models. Hope this further clears your concern on the larger models.
> > >
> > >
> > > Regarding Q5: Since our method directly modifies the model weights without any data, it needs specific designs for specific architectures. Yet as we mentioned in the previous response: as long as we can isolate a certain path in the model, we can deploy our DFBA with theoretical guarantees. We are sorry that given the short response time, we cannot quickly adapt DFBA to other architectures and give you further results but we will keep that as our future work.
> > >
> > > Nevertheless,  DFBA can achieve nearly 100% ASR in less than 1 second of injection time, with no data and minimal impact on the model's original performance. We believe this somewhat compensates for the current limitations of DFBA in terms of extensibility.
> > >
> > > Once again, we thank the reviewer for their profound insights and productive discussion, as well as for their support and recognition of our paper!

---

### Author Rebuttal · Authors · 2024-08-07

We thank all the reviewers for your valuable comments!

Here we address some common questions for all reviewers:

**CQ1: Maintaining method stability and Backdoor Accuracy.**

**CA1:** We discuss DFBA's performance in two aspects: Attack Success Rate (ASR) and Backdoor Accuracy (BA).

Regarding ASR, our method is stable because we technically eliminate interference from other neurons on our implanted backdoor path, ensuring the trigger always activates the backdoor path. By controlling the amplification factor $\lambda\gamma^{L-1}$, we can set the target class's confidence score to a large number when the backdoor path is activated, guaranteeing nearly 100% ASR.

For BA, potential instability mainly comes from randomly selecting neurons, which may change the model's original performance. However, according to Theorem 1 in our paper, this change is equivalent to pruning one neuron in each of the L-1 layers, which usually has a small impact on model performance. Some pruning methods, such as [1], even show that ResNet and VGG can maintain model performance almost unchanged after pruning about 70% of the parameters. We have also experimentally proven this:

We repeated experiments 5 times with different random seeds on CIFAR10+ResNet-18 and GTSRB+VGG16. The BA was $91.46 \pm 0.32$ and $95.48 \pm 0.46$ respectively, which is almost identical to the results we reported in our paper, and ASR was 100% for both. We will repeat all major experiments and report errors in the future.

Additionally, we can introduce some Data-free pruning methods [2,3] to enhance DFBA's stability in BA without compromising our threat model and theoretical guarantees. If the threat model is relaxed to allow access to some data, we can simply extend our method to only modify neurons that are not activated on most data. In this case, our method remains efficient (taking less than 1s), stable (almost always achieving 100% ASR), and cannot be removed by various fine-tuning methods or gradient-based methods (such as NC).

**CQ2: Adaptive defense measures.**

**CA2:** We designed two adaptive defense methods tailored for DFBA. These methods exploit the fact that our DFBA-constructed backdoor paths are rarely activated on clean data and that some weights are replaced with zeros when modifying the model weights:

1. Anomaly detection: Check the number of zero weights in the model.
2. Activation detection: Remove neurons in the first layer that always have zero activation values on clean datasets.

To counter these adaptive defenses, we replaced zero weights with small random values. We used Gaussian noise with $\sigma=0.001$. We conducted experiments on CIFAR10 with ResNet-18, using the default hyperparameters from the paper. Results show we still achieve 100% ASR with less than 1% performance degradation.

This setup eliminates zero weights, rendering anomaly detection ineffective. We also analyzed the average activation values of 64 filters in the first layer on the training set (see Figure in PDF). Our backdoor path activations are non-zero and exceed many other neurons, making activation detection ineffective.

We tested fine-pruning and Neural-Cleanse (Anomaly Index = 1.138) under this setting. Both defenses failed to detect the backdoor.

We didn't adopt this setting in the paper as it compromises our theoretical guarantees. Our goal was to prove the feasibility and theoretical basis of a novel attack method. Additionally, we can distribute the constructed backdoor path across multiple paths to enhance robustness. We plan to discuss these potential methods in the next version.

**CQ3: Comparison with Lv et al. [4].**

**CA3:** Lv et al. [4] also proposed a data-free backdoor injection method. They fine-tune the model using a substitute dataset to inject the backdoor and design a loss function to maintain the target model's performance on the original task. Our DFBA differs from [4] in several ways:

1. **Definition of "data-free", and the method of injecting backdoor are different**: [4] require a substitute dataset for fine-tuning, while DFBA injects backdoors by directly modifying model weights without any data.

2. Theoretical guarantees: Our method provides formal guarantees for backdoor ASR and robustness against certain defense methods.

3. Higher ASR and stealthiness for small triggers: In experiments with CIFAR10 and ResNet-18 using a $3 \times 3$ trigger, [4] achieved 30.34% ASR, while DFBA achieved 100% ASR. According to Table 7 in [4], their backdoor is detectable by Neural Cleanse when the trigger size is smaller than $6 \times 6$. We confirmed this (Anomaly Index=3.14) in our experiments with a $3 \times 3$ trigger, while DFBA remained undetected (Anomaly Index=0.94) under the same conditions.

4. Faster injection: On an NVIDIA A100 GPU, DFBA injects backdoors in 0.0654 seconds for ResNet-18 model trained on CIFAR10, and 0.0733 seconds for ResNet-101 trained on ImageNet. In contrast, methods like [4] require over 5 minutes for ResNet-18 on CIFAR10 and over 50 minutes for VGG16 on ImageNet.

However, since [4] inject the backdoor by fine-tuning the model using a substitute dataset, this allows their method to be easily applied to various models after constructing poisoning datasets. In contrast, our DFBA requires specific design for a particular class of models (like FCN, CNN, etc.).

We will conduct more comprehensive comparisons with [4] and include them in the paper.


We kindly request that you inform us of any remaining ambiguities or concerns. We are more than willing to address additional questions and conduct further experiments should the reviewers deem it necessary.

[1] Lin M, Ji R, Zhang Y, et al. Channel pruning via automatic structure search

[2] Srinivas S, Babu R V. Data-free parameter pruning for deep neural networks

[3] Kim W, Kim S, Park M, et al. Neuron merging: Compensating for pruned neurons

[4] Lv P, Yue C, Liang R, et al. A data-free backdoor injection approach in neural networks

---

### Decision · Program_Chairs · 2024-09-25

**Decision:**

Accept (poster)

**Comment:**

The paper proposes a backdoor attack based on adjusting the weight parameters of a neural network (i.e. model poisoning). It does not require knowledge of the training data, but just access to the pre-trained model. The attack is inspired by a previous paper at NeurIPS '22 (referred to as [27] in the paper), which it compares against, showing better performance. I retain the concerns expressed by reviewer Npgg, namely, that a consistent ASR of 100% may be due to a limited or biased evaluation, which is far less extensive than the experiments reported in a recent (competing) USENIX '23 publication (mentioned by reviewer Npgg). However, this work also provides some interesting contributions, and considering the positive feedback from all reviewers, I'm leaning toward accepting it.